# A method for Boolean analysis of protein interactions at a molecular level

Doroteya Raykova [1,8] ✉, Despoina Kermpatsou [1,8], Tony Malmqvist [2], Philip J. Harrison[1], Marie Rubin Sander [1], Christiane Stiller [1], Johan Heldin [1], Mattias Leino[1], Sara Ricardo [3,4,5], Anna Klemm [6], Leonor David [3,4], Ola Spjuth [1], Kalyani Vemuri [7], Anna Dimberg [7], Anders Sundqvist[1], Maria Norlin [1], Axel Klaesson[1], Caroline Kampf[2] & Ola Söderberg [1] ✉

Determining the levels of protein–protein interactions is essential for the analysis of signaling within the cell, characterization of mutation effects, protein function and activation in health and disease, among others. Herein, we describe MolBoolean – a method to detect interactions between endogenous proteins in various subcellular compartments, utilizing antibody-DNA conjugates for identification and signal amplification. In contrast to proximity ligation assays, MolBoolean simultaneously indicates the relative abundances of protein A and B not interacting with each other, as well as the pool of A and B proteins that are proximal enough to be considered an AB complex. MolBoolean is applicable both in fixed cells and tissue sections. The specific and quantifiable data that the method generates provide opportunities for both diagnostic use and medical research.

Proteomics-based approaches have proven essential in various research settings for purposes such as detection of markers in cancer diagnostics, understanding fundamental research questions like signal transduction mechanisms, regulation of gene expression and mutation effects, identification of vaccine targets, elucidation of the mechanisms of drug action, etc. Over the years, a plethora of such methods to suit the complexity and diversity of research questions has been developed. Several of them are based on genetic constructs, where candidate proteins are fused with reporter molecules that upon interaction reconstitute a functional reporter (e.g., yeast two-hybrid[1], mammalian membrane two-hybrid[2], and bimolecular fluorescence complementation[3]). Alternatively, Förster resonance energy transfer (FRET) can be used to determine proximal binding of fluorophores, with a concomitant change in emission spectra/lifetime[4]. More specifically, FRET is based on the transfer of energy between light-sensitive molecules—a donor and an acceptor, which has an absorption

spectrum overlapping with the emission spectrum of the donor. The efficiency of the resonance energy transfer is strongly dependent on the distance between the fluorophores[5]. While FRET is a sensitive technique suitable for determining intermolecular proximity in the range of 1–10 nm[6], among its limitations are the low signal-to-noise ratio[7] and the necessity to fuse the target proteins to the acceptor/donor, which makes the method unfit for clinical use. An additional consideration to keep in mind is that the distance between fluorophores is not necessarily identical to that of the target proteins.

To determine interactions between native proteins, most methods rely on antibodies conjugated to functional groups, for example, antibody-based FRET[8], in situ proximity ligation assay (in situ PLA)[9,10], or proximity-dependent initiation of hybridization chain reaction (proxHCR)[11]. Both in situ PLA and proxHCR rely on dual-target recognition with secondary antibodies conjugated to oligonucleotides (so-called proximity probes), and utilize DNA as a reporter of proximity

[1]Department of Pharmaceutical Biosciences, Science for Life Laboratory, Uppsala University, Biomedical center, SE-751 24 Uppsala, Sweden. [2]Atlas Antibodies AB, Bromma, Sweden. [3]Faculty of Medicine, University of Porto, Porto, Portugal. [4]Differentiation and Cancer Group, Institute for Research and Innovation in Health (i3S) of the University of Porto/Institute of Molecular Pathology and Immunology of the University of Porto (Ipatimup), Porto, Portugal. [5]Department of Sciences, University Institute of Health Sciences (IUCS), CESPU, Gandra, Portugal. [6]Vi2, Department of Information Technology and SciLifeLab BioImage Informatics Facility, Uppsala University, Uppsala, Sweden. [7]Department of Immunology, Genetics and Pathology, Uppsala University, Rudbeck Laboratory, Uppsala, Sweden. [8]These authors contributed equally: Doroteya Raykova, Despoina Kermpatsou. ✉e-mail: doroteya.raykova@farmbio.uu.se; ola.soderberg@farmbio.uu.se

events, which allows for powerful signal amplification and improved signal-to-noise ratio over traditional FRET. It is important to emphasize that what all of the above-mentioned methods detect is proximity between proteins. For in situ PLA, the proximity threshold is determined by the antibody size and the oligonucleotide length of the probes. The hybridization of a pair of circularization oligos to the probes, resulting in the creation of a circular ligation product, is only possible when the attachment points of the oligonucleotide components of the probes are located within FRET range (below 10 nm)[6]. When primary antibodies are conjugated to the oligonucleotides (i.e., in the case of primary probes) the maximal theoretical distance between targeted epitopes is 30 nm[9], while for secondary proximity probes it is estimated to be 40 nm. However, highly expressed proteins may be localized very close to each other−less than 40 nm apart−even if they do not interact. In order to confidently interpret data generated with such methods, it is crucial to obtain information not only on the number of proximity events, but also on the amounts of free proteins involved that can be used to normalize data. To be able to detect both the proteins in complex and the pool of non-interacting proteins, we developed a method−MolBoolean−which is based on the Boolean operators NOT and AND on a molecular level. It reports the amounts of protein A and protein B that do not participate in an interaction with each other (NOT), while at the same time also visualizes the pool of A and B proteins that are proximal enough to be considered an AB complex (AND).

In this work, we first demonstrate the utilization of MolBoolean for specific dual detection of single proteins, using a pair of antibodies targeting one protein. To establish the applicability of MolBoolean for simultaneous detection and quantification of free proteins and proteins in complex, we then investigate multiple established protein−protein interactions described in literature. In order to highlight our method's versatility, we perform stainings against proteins localized in various cell compartments, as well as observe changes in free and complex-bound states upon ligand stimulation, siRNA silencing, and other conditions vs no-treatment conditions. Further, we validate the use of MolBoolean not only in fixed cells, but also in different types of formalin-fixed paraffin-embedded (FFPE) tissue sections. In parallel to the MolBoolean analyses, we perform well-established classic techniques not only to validate our findings, but also to provide grounds to compare and contrast the merits of MolBoolean with those of in situ PLA (visualizing protein interactions) and immunofluorescence (IF, suitable also for visualization of free proteins and compartmentalization). We herein demonstrate that MolBoolean provides information on protein interaction in a manner similar to in situ PLA (i.e., by detecting discrete rolling circle amplification products (RCPs)), but also on the relative quantities of individual proteins, which allows for quantification of the RCPs in each category on a single cell level.

## Results

### Principle of the MolBoolean method
Similar to in situ PLA, MolBoolean, too, relies on the use of proximity probes and rolling circle amplification (RCA) as means for generating and amplifying signal. However, MolBoolean uses a preformed DNA circle as information receiver that would later indicate whether one or two proteins have been detected. At the basis of MolBoolean, like other immunoassays, is the specific binding of a pair of antibodies to their respective protein targets, and the subsequent recognition of each primary antibody by a proximity probe. The MolBoolean proximity probe is essentially a secondary antibody conjugated to a DNA oligonucleotide, termed "arm", which is complementary to a specific region in the aforementioned circle. Whenever the information receiver circle and the complementary region in a proximity probe hybridize (Fig. 1a), double-stranded DNA is formed that can be recognized by a nickase. A key feature of this enzyme is its ability to recognize double-stranded

DNA motifs and cut just one of the strands in a defined position. Therefore, once the recognition sequence is formed, the nickase creates a nick in the circle, but not in the proximity probe (Fig. 1a, b, nick position indicated by cyan arrowhead). The circle is at this point interrupted by either one or two nicks, depending on whether one or two proximity probes have bound to their complementary regions in it. Next, one or two identifier "tag" oligonucleotides specific for their respective proximity probe get incorporated in the DNA circle by virtue of their complementarity to a loop-and-hairpin region in the probe (Fig. 1c), and the circle is sealed whole by ligation (Fig. 1d). Consequently, the re-formed circle now contains information on whether it has interacted with one or two proximity probes. The circle then gets amplified via RCA, forming long concatemeric DNA products (Fig. 1e). These RCPs are then hybridized with fluorophore-labeled tag-specific detection oligonucleotides to differentially visualize the identities of the incorporated tags (Fig. 1f). Single-labeled RCPs represent free proteins, while dual-stained ones represent interactions.

Detailed information on how all oligonucleotides were designed and the significance of all their functional regions for MolBoolean is available in Supplementary Notes, subsection Oligonucleotide Design.

### In solution testing of the MolBoolean specificity
To validate the specificity of the enzymatic steps in the MolBoolean method, we performed in solution tests to ensure that hybridization of the arms to the circle provided a template for the nickase, and that the tag oligonucleotides were specifically incorporated in the nicked DNA circle (Supplementary Fig. 1). The circle was only nicked when an arm oligonucleotide was added, showing that the nickase requires a double-stranded DNA substrate (Supplementary Fig. 1, wells 6–9). Tags were only incorporated where the cognate arm was hybridized to the circle, which demonstrates that tag incorporation is dependent on the identity of the proximity probes (Supplementary Fig. 1, wells 10–14). In contrast to the in situ protocol (see Methods) where rigorous washes were used to remove all enzymes before the next step, the experiments in solution included heat inactivation steps instead. Heating leads to partial or complete denaturation of the double-stranded oligonucleotide complexes. Even though they can reform after cooling of the sample, that lowers the efficiency of hybridization compared to in situ conditions. Therefore, the in solution test was used to validate specificity rather than efficiency.

### Sensitive single protein detection
MolBoolean is designed to detect free and bound proteins alike. To test the method's performance in conditions in which all probes were bound in proximity (provided the specificity of all antibodies used is 100%), we co-stained cells with two antibodies against distinct epitopes of a protein. The principle is analogous to how ELISA, PLA and proximity extension assay (PEA) achieve their high selectivity – via dual recognition of a single protein by two pairs of antibodies, – and the first bottleneck is the same: the specificity of the primary antibodies used. This assay demonstrates how MolBoolean can be applied for antibody validation, showing whether two antibodies against the same target protein are equally good at binding it, and the extent to which they cross-react with other proteins. Figure 2 demonstrates MolBoolean staining with two antibodies against β-catenin raised in different species with a high degree of overlap in the honeycomb pattern typical of β-catenin[12], and compares the results with in situ PLA and IF performed with the same primary antibodies. Omitting controls in which either one of the antibodies was excluded are shown both for MolBoolean and for in situ PLA.

### MolBoolean analysis of E-cadherin and β-catenin
To validate the performance of the MolBoolean method we performed a series of stainings for E-cadherin and β-catenin in various conditions. E-cadherin is a cell adhesion protein that primarily localizes in the

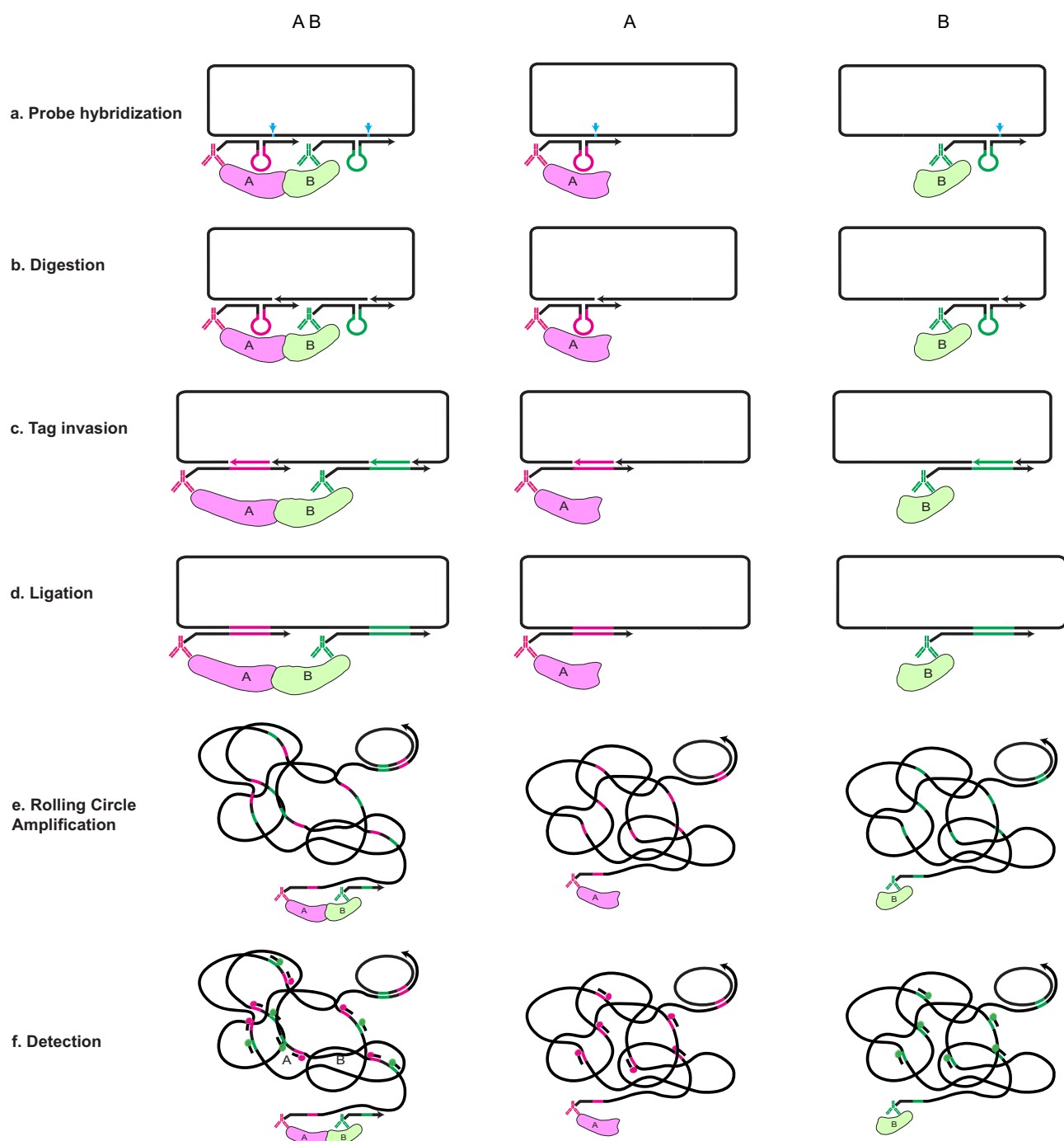

**Fig. 1 | Schematic representation of the MolBoolean principle for detection of interacting and free proteins A and B. a** After binding their respective target proteins A and B, proximity probes A (black and magenta) and B (black and green) hybridize to the circle. Arrows signify oligonucleotide polarity. **b** The circle gets enzymatically nicked (cyan arrowhead indicates nicking position). **c** The circle gets invaded by reporter tags (tag A in magenta, tag B in green). **d** Enzymatic ligation of the reporter tags to the circle follows. **e** Rolling circle amplification (RCA) creates long concatemeric products (RCPs). **f** RCPs are detected via fluorescently labeled tag-specific detection oligonucleotides.

membrane but can also be found in the endosomes and the trans-Golgi network[13] (Uniprot P12830[14]). β-catenin is a transcriptional coactivator in the Wnt signaling pathway, and plays an important role in cell adhesion together with E-cadherin (Uniprot P35222[14]). Upon phosphorylation, β-catenin accumulates in the nucleus; when interacting with E-cadherin, it is found in the plasma membrane[15]. We began with a staining in MCF7 cells, which express both E-cadherin and β-catenin, against a biological control (U2OS cells) that does not express E-cadherin[16]. As expected, in MCF7 cells we observed free proteins in the cytoplasm, as well as the specific honeycomb pattern of interaction

in the plasma membrane, reflecting colocalization of the two proteins in the adherens junctions (Fig. 3a, expanded in Supplementary Fig. 2a). In contrast, in the osteosarcoma cell line U2OS only free β-catenin was detected (Fig. 3a, expanded in Supplementary Fig. 2a).

To further test the robustness of our method, we performed a no-interaction control stain, which targeted E-cadherin and Lamin A/C (LMNA/C). Lamin A/C, in contrast to β-catenin, primarily localizes in a different subcellular compartment than E-cadherin. Lamin A/C is part of the nuclear lamina[17], therefore little to no interaction should occur between the two proteins. Both MolBoolean and in situ PLA detected a

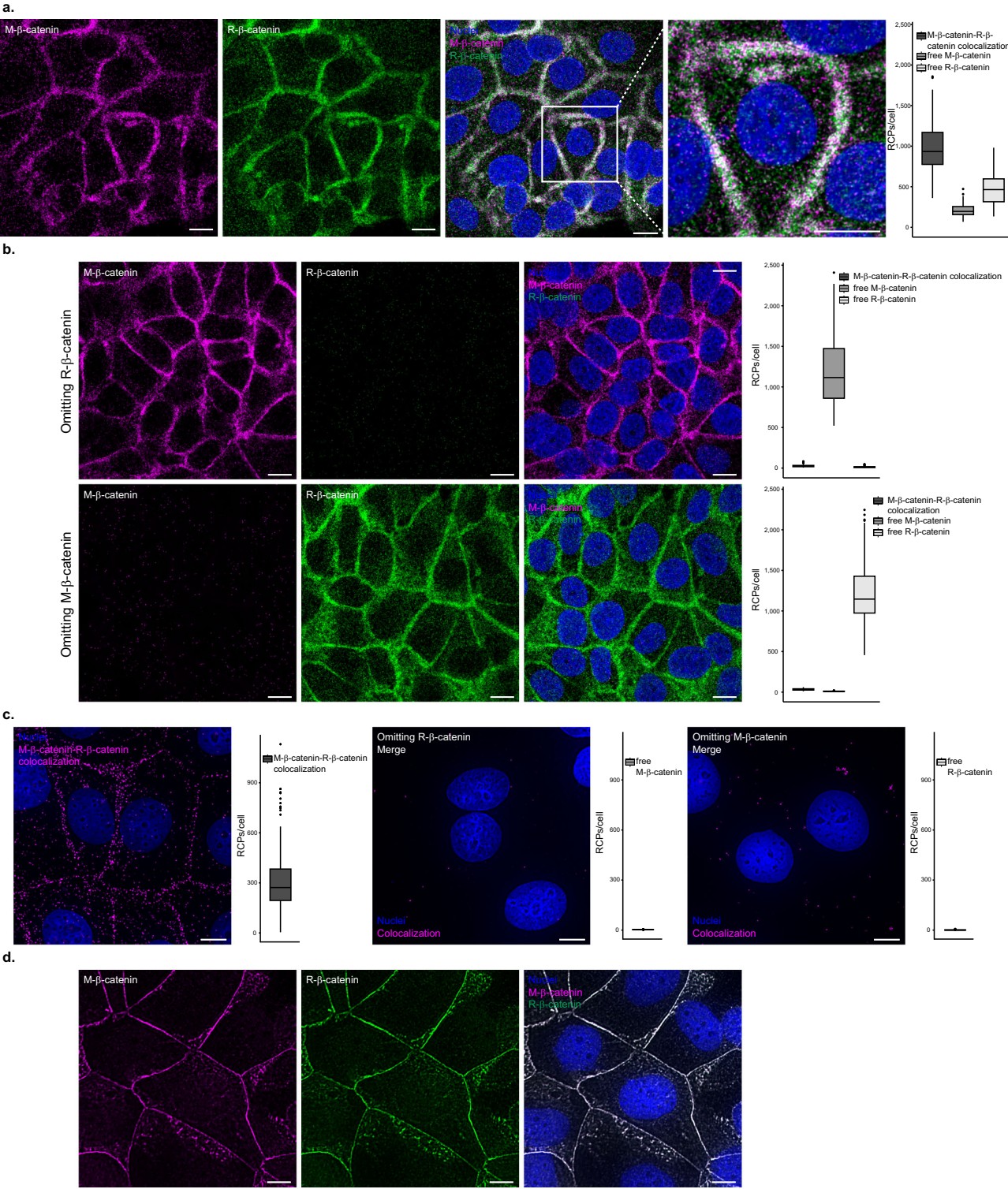

**Fig. 2 | Single protein targeting and antibody validation. a** MolBoolean staining and quantification with two antibodies, one raised in mouse (M, magenta), and the other raised in rabbit (R, green), against distinct epitopes of β-catenin in MCF7 cells. Dual staining is shown in white and Hoechst33342 staining of nuclei is shown in blue. **b** MolBoolean technical controls, in which either one of the primary antibodies was omitted from the reaction mix. **c** In situ PLA colocalization staining, omitting controls and quantifications with the same pair of anti-β-catenin antibodies in MCF7 cells. In situ PLA signals are shown in magenta and nuclei in blue. **d** Immunofluorescent staining of MCF7 cells with the same pair of anti-β-catenin

antibodies. Mouse antibody (magenta), rabbit antibody (green), overlay (white) and nuclei (blue). White frames depict an area shown in enlarged view in the following panel. Scale bars = 10 μm. Quantification of protein complexes and free proteins (MolBoolean) or protein complexes only (in situ PLA) shown as number of RCPs per cell. Data pooled from three independent experiments. Box plots show median, Q1 to Q3 range, lower and upper whiskers at maximum 1.5 times the interquartile range. Outliers shown as solid circles. Source data are provided as a Source Data file.

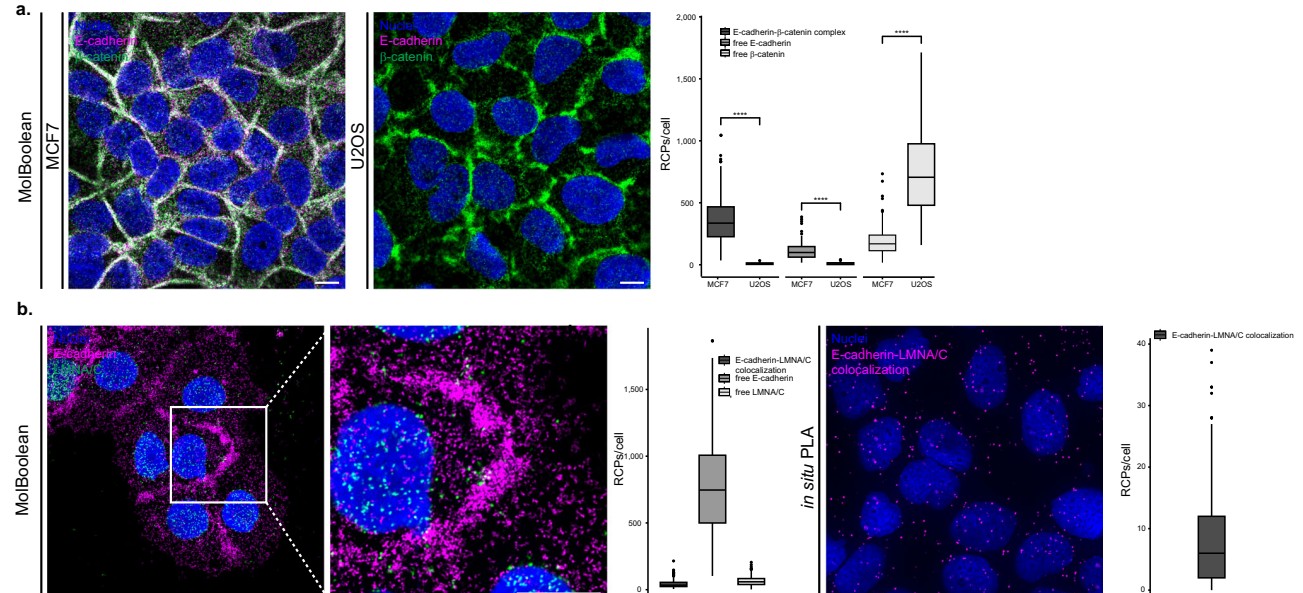

**Fig. 3 | MolBoolean staining of E-cadherin with an interaction partner vs no-interaction partner. a** Co-stain of E-cadherin which is differentially expressed in MCF7 (positive for E-cadherin) and U2OS cells (negative for E-cadherin) and interaction partner β-catenin. ($p = 1.76e-55$; $p = 1.89e-55$; $p = 7.84e-52$ for E-cadherin-β-catenin complex, free E-cadherin, free β-catenin respectively). MolBoolean signals are shown for E-cadherin (magenta), β-catenin (green), E-cadherin-β-catenin complex (white) and nuclei (blue). **b** E-cadherin and LMNA/C in HaCaT cells are expected not to colocalize. MolBoolean signals are shown for E-cadherin (magenta), LMNA/C (green), E-cadherin-LMNA/C complex (white) and nuclei (blue). In situ PLA signals are shown in magenta and nuclei in blue. White frames depict an area shown in enlarged view in the following panel. Scale bars = 10 μm. Quantification of protein complexes and free proteins (MolBoolean) or protein complexes only (in situ PLA) shown as number of RCPs per cell. In (**a**) $n_{MCF7} = 243$, $n_{U2OS} = 125$ cells. Data pooled from three independent experiments. Kruskal–Wallis and two-sided Dunn's test with Bonferroni correction was used to analyze statistical variance. Box plots show median, Q1 to Q3 range, lower and upper whiskers at maximum 1.5 times the interquartile range. Outliers shown as solid circles. ****$p < 0.0001$. Source data are provided as a Source Data file.

small number of interactions, reflecting spurious antibody binding events (Fig. 3b). In addition, MolBoolean recorded high expression of free E-cadherin and low expression of free LMNA/C. Omitting controls where either one of the primary antibodies was not used, and IF (Supplementary Fig. 2b) were performed in parallel and resulted in staining consistent with MolBoolean. For an additional no-interaction control involving another pair of target proteins in different organelles, see Supplementary Notes and Supplementary Fig. 2c, d.

To validate that the dual-colored RCA products observed in E-cadherin−β-catenin stains contain both tags and are not just individual single-colored RCPs located in close proximity, we designed padlock probes targeting the MolBoolean proximity probes (see Methods for design). Padlock probes consist of two target-complementary segments, connected by a linker sequence[18]. Upon recognition of the target DNA sequence (Fig. 4a), the 5′ and the 3′ end of the padlock probe can be joined by ligation with T4 ligase (Fig. 4b), creating a circular DNA molecule that is amplifiable by RCA[18] (Fig. 4c). As a result, regardless of the target proteins' proximity, each padlock probe always generates its own individual fluorescent signal and never a dual signal (Fig. 4d). We therefore used primary antibodies against E-cadherin and β-catenin in MCF7 cells and then either performed the MolBoolean protocol, or substituted the circle hybridization, nicking and tag ligation steps with padlock probe hybridization (Fig. 4e). Due to the higher overall number of signals detected with padlock probes, data were normalized by dividing the signals in each category by the total number of RCPs per cell. Quantification, normalization, and comparison of the resulting images showed a significantly higher fraction of complexes per cell in the MolBoolean *versus* the padlock experiment.

Next, to validate that the MolBoolean method can specifically visualize interacting proteins, we assayed for E-cadherin and β-catenin in cells that harbor a pathogenic cytoplasmic missense mutation (V832M) in the β-catenin binding site of E-cadherin, which leads to reduced interaction, lowered surface expression of E-cadherin and a

failure of mutants to aggregate in culture[19]. By using a pair of AGS cell lines stably transfected with either wild-type (WT) E-cadherin or V832M, we reasoned that the expression levels of E-cadherin would be comparable, but the levels of interaction should differ[19]. In agreement with previous in situ PLA data[19], in mutant cells we observed decreased cell aggregation and a dramatic reduction in complex formation, while the levels of free E-cadherin remained stable (Fig. 5a).

As a demonstration of MolBoolean's ability to detect dynamic changes in protein complex formation under different conditions, we once again resorted to the hallmark interaction of cell adhesion, E-cadherin−β-catenin, but this time included a cell treatment regime (Fig. 5b and Supplementary Fig. 3a). Prolonged treatment with TGF-β1 has been shown to lead to loss of cell-cell contacts, disruption of the interaction between E-cadherin and β-catenin in the adherens junctions, and subsequent E-cadherin translocation to the cytoplasm[20]. We therefore analyzed the pools of free and bound E-cadherin and β-catenin in HaCaT cells before ("control" condition) and after 48 h of TGF-β1 treatment ("TGF-β1 treated" condition) in order to assess how MolBoolean performs in an inducible biological system. Both Mol-Boolean (Fig. 5b) and IF staining (Supplementary Fig. 3a) showed morphological changes and disruption of cell-cell contacts in the treated cells. To account for the increased cell area and consequent increase in the total number of RCPs/cell detected in the treated condition, we normalized all MolBoolean data by dividing the number of signals in each category (free E-cadherin, free β-catenin, or proteins in complex) over the total number of detected signals in each cell (Fig. 5b, MolBoolean quantification). We observed a significant decrease in the fraction of E-cadherin−β-catenin complexes and a significant increase in the free E-cadherin relocating from the adherens junctions to the cytoplasm as a result of the TGF-β1 treatment, whereas the levels of unbound β-catenin remained stable (Fig. 5b, MolBoolean quantification). Since there is only one category of signal detected by in situ PLA, i.e., interactions, normalization cannot be performed for

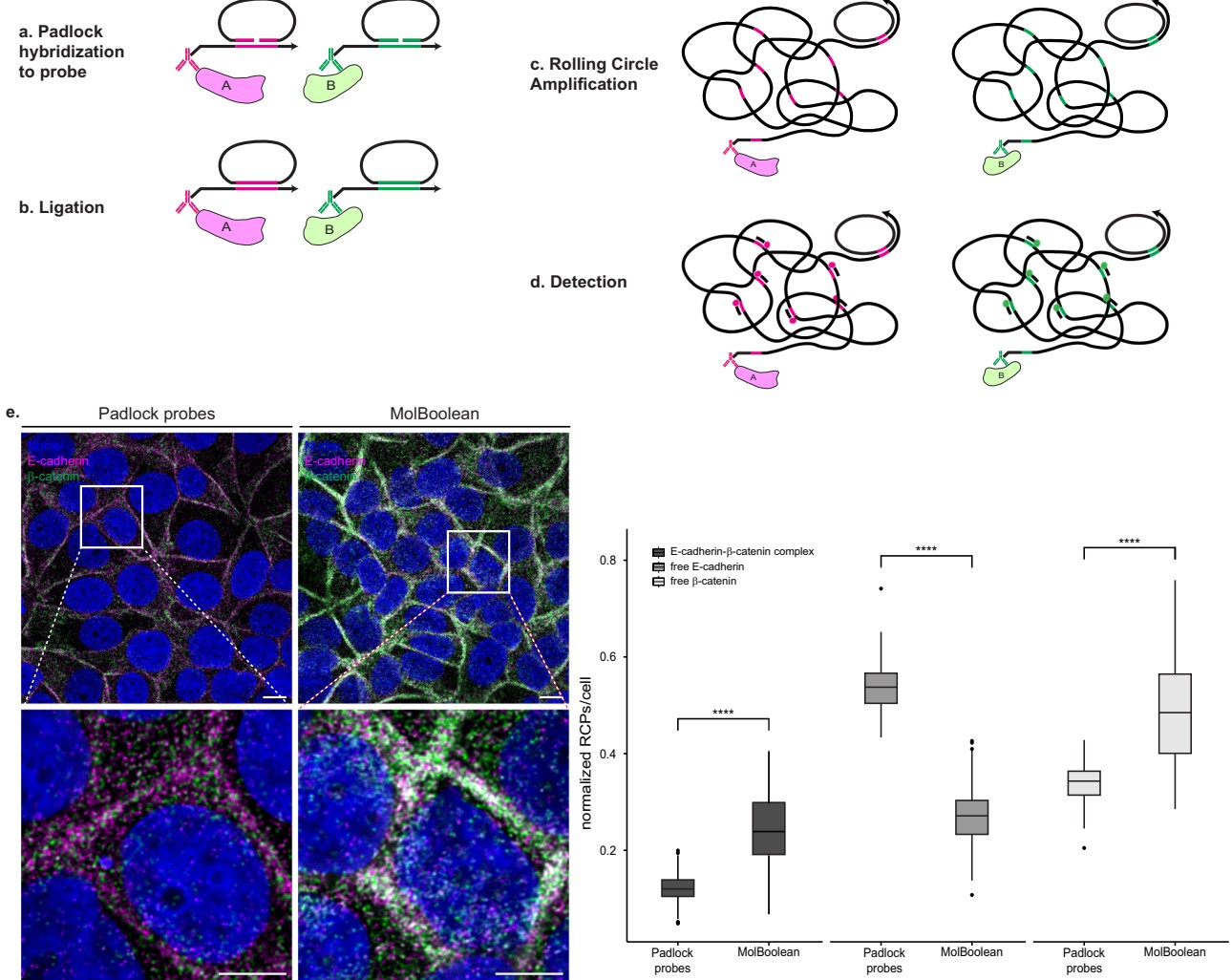

**Fig. 4 | Padlock probe design and in situ application in fixed cells. a** After proximity probes A and B bind their respective target proteins A and B, padlock oligonucleotides A and B hybridize to their respective arm in place of the Mol-Boolean circle. Each padlock contains the complementary tag sequence (magenta corresponds to tag A and green corresponds to tag B). **b** Enzymatic ligation of the 5' and 3' ends of each padlock, templated by the corresponding arm, leads to padlock circularization. **c** RCA, primed by the arms, creates long concatameric RCPs. **d** RCPs are detected via fluorescently labeled tag-specific detection oligonucleotides. **e** In situ application of the padlock probes and signal quantification. E-cadherin and β-catenin co-stain in MCF7 cells with padlock probes vs MolBoolean. ($p = 1,46\text{e}-57$; $p = 9,69\text{e}-68$; $p = 1,98\text{e}-50$ for E-cadherin-β-catenin complex, free E-cadherin, free β-catenin respectively). E-cadherin is shown in magenta, β-catenin in green, E-cadherin-β-catenin complex (MolBoolean), or overlay (padlock probes) in white and nuclei in blue. White frames depict an area shown in enlarged view in the following panel. Scale bars = 10 μm. Quantification of protein complexes and free proteins shown as normalized number of RCPs per cell. $n_{\text{padlock}} = 173$, $n_{\text{MolBoolean}} = 243$ cells; data pooled from three independent experiments. Kruskal–Wallis and two-sided Dunn's test with Bonferroni correction was used to analyze statistical variance. Box plots show median, Q1 to Q3 range, lower and upper whiskers at maximum 1.5 times the interquartile range. Outliers shown as solid circles. ****$p < 0.0001$. Source data are provided as a Source Data file.

in situ PLA data. In situ PLA showed an approximately twofold increase in interaction for the TGF-β1 treated cells compared to control (Fig. 5b, in situ PLA quantification). In addition, we performed the same experiment by using primary probes, where E-cadherin and β-catenin primary antibodies were directly conjugated to the arms, and obtained comparable results (see Supplementary Notes and Supplementary Fig. 3b).

Finally, we explored the E-cadherin–β-catenin interaction in FFPE prostate tissue, and observed honeycomb pattern of staining (Fig. 5c and Supplementary Fig. 3c) with MolBoolean, in situ PLA and IF. For MolBoolean, nearly 50% of the detected signal per image was from E-cadherin–β-catenin complexes (Fig. 5c, pie chart).

**MolBoolean analysis of proteins in various cell compartments**
To showcase the MolBoolean performance for various biological targets, we assayed several established protein–protein interactions in different cell organelles and went on to compare the results to in situ PLA (to ensure that protein–protein interactions are detected) and IF (to show that the staining patterns of the individual proteins are comparable). Technical controls for in situ PLA and MolBoolean in which one or the other primary antibody of the pair was omitted are shown in Supplementary Information for each experiment.

To demonstrate that MolBoolean can be utilized for the quantification of free and interacting proteins confined to "crowded" compartments of the cell, we applied our method to stain MCF7 cells for Emerin (EMD) and Lamin B1 (LMNB1) (Fig. 6a and Supplementary Fig. 4a). Lamin B1 is localized in the nuclear membrane and to some extent the nucleoplasm, and, like other lamins, provides structure to the nuclear lamina and participates in multiple nuclear processes[21,22]. Emerin, a stabilizer of the nuclear envelope found on the inner and outer nuclear membrane as well as on the membrane of the endoplasmic reticulum (ER), is a known interaction partner to Lamin B1[23,24].

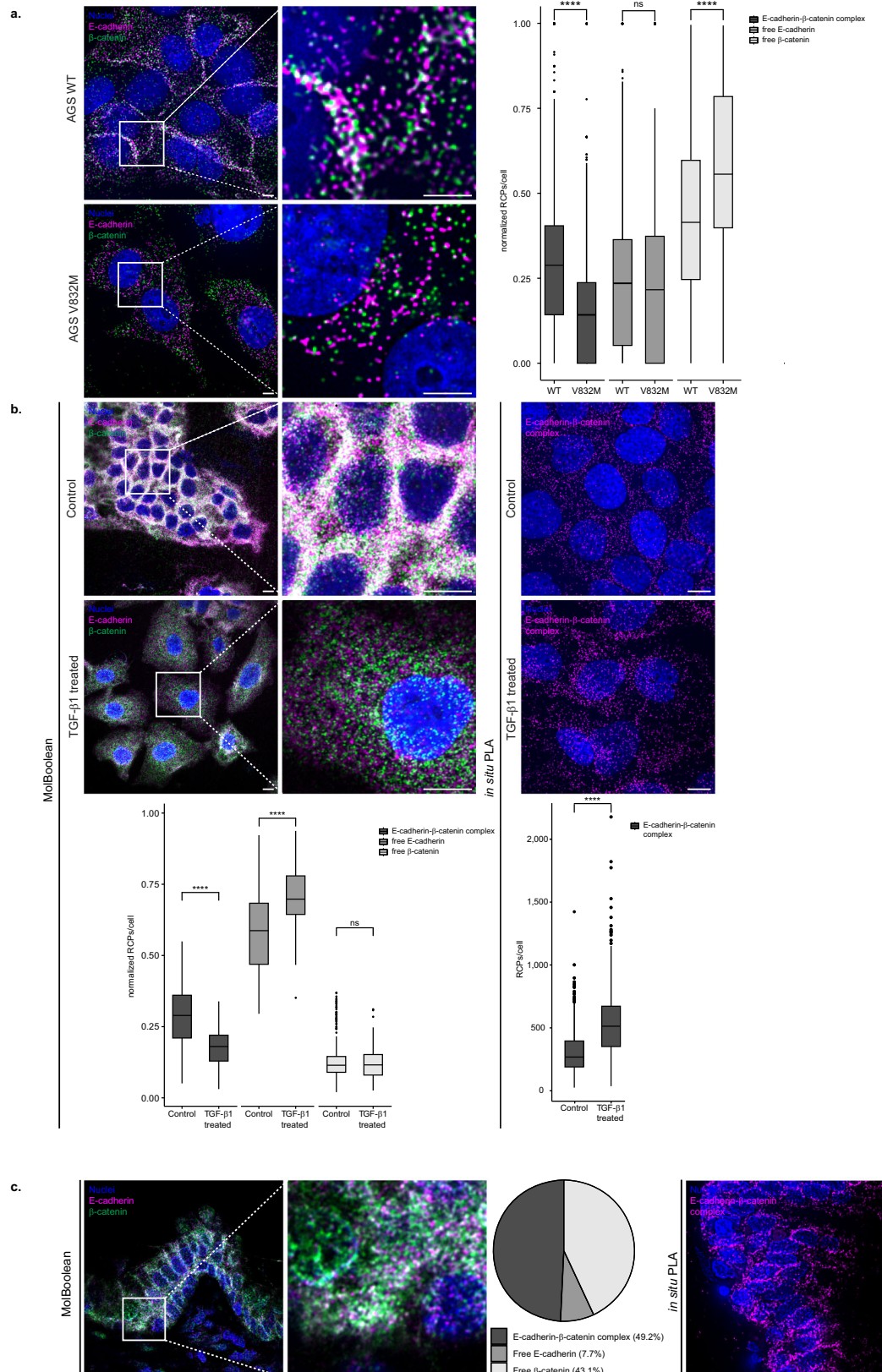

MolBoolean staining demonstrated a fraction of free Emerin in the region of the ER, a fraction of non-interacting Lamin B1 in the nucleoplasm, and a significant region of interactions in the nuclear envelope (Fig. 6a). The presence of EMD-LMNB1 complexes was verified by in situ PLA (Fig. 6a, right panel), and the nuclear co-localization of the two proteins was additionally shown with IF (Supplementary

Fig. 4a, bottom panel). Technical controls for MolBoolean and in situ PLA in which either the anti-Emerin, or the anti-Lamin B1 antibody was omitted (omitting controls) are displayed in Supplementary Fig. 4a.

Nuclear proteins are notoriously difficult to stain, so we tested MolBoolean on an assay that features nuclear interactions: FUS-HNRNPM (Fig. 6b). FUS is a DNA/RNA-binding protein residing

**Fig. 5 | MolBoolean and in situ PLA staining of E-cadherin and β-catenin under various conditions in fixed cells or in tissues. a** Co-stain in stable AGS cell clones transfected with wild-type E-cadherin (WT, top panel) or E-cadherin with a V832M mutation in the β-catenin binding site (AGS V832M, bottom panel). (p = 6.51e−48; p = 0.17; p = 4.53e−26 for E-cadherin-β-catenin complex, free E-cadherin, free β-catenin respectively). **b** Co-stain in HaCaT cells, in the absence ("control", top) or presence ("TGF-β1 treated", bottom) of TGF-β1. (p = 7.64e−24; p = 1.66e−15; p = 0.92 E-cadherin-β-catenin complex, free E-cadherin, free β-catenin respectively (Mol-Boolean) and p = 2,94e−55 (in situ PLA)). **c** Co-stain in FFPE kidney tissue sections. MolBoolean signals are shown for E-cadherin (magenta), β-catenin (green), E-cadherin-β-catenin complex (white), and nuclei (blue). In situ PLA signals are shown in magenta and nuclei in blue. White frames depict an area shown in enlarged view

in the following panel. Scale bars = 10 µm. Quantification of protein complexes and free proteins (MolBoolean) or protein complexes only (in situ PLA) shown as number of RCPs per cell in the case of fixed cell analysis, or in percentage of RCPs in each category (free protein A, free protein B, and AB complex) per frame in the case of tissue analysis. $n_{WT}$ = 1160, $n_{V832M}$ = 810 cells (**a**), and $n_{control}$ = 371, $n_{treated}$ = 113 cells (**b**). Data pooled from three independent experiments, and in (**a**, **b**) normalized against total number of signal/cell. Two-sided Wilcoxon rank sum test was used to analyze statistical variance in fixed cell data. Box plots show median, Q1 to Q3 range, lower and upper whiskers at maximum 1.5 times the interquartile range. Outliers shown as solid circles. ****p < 0.0001, ns not significant. Source data are provided as a Source Data file.

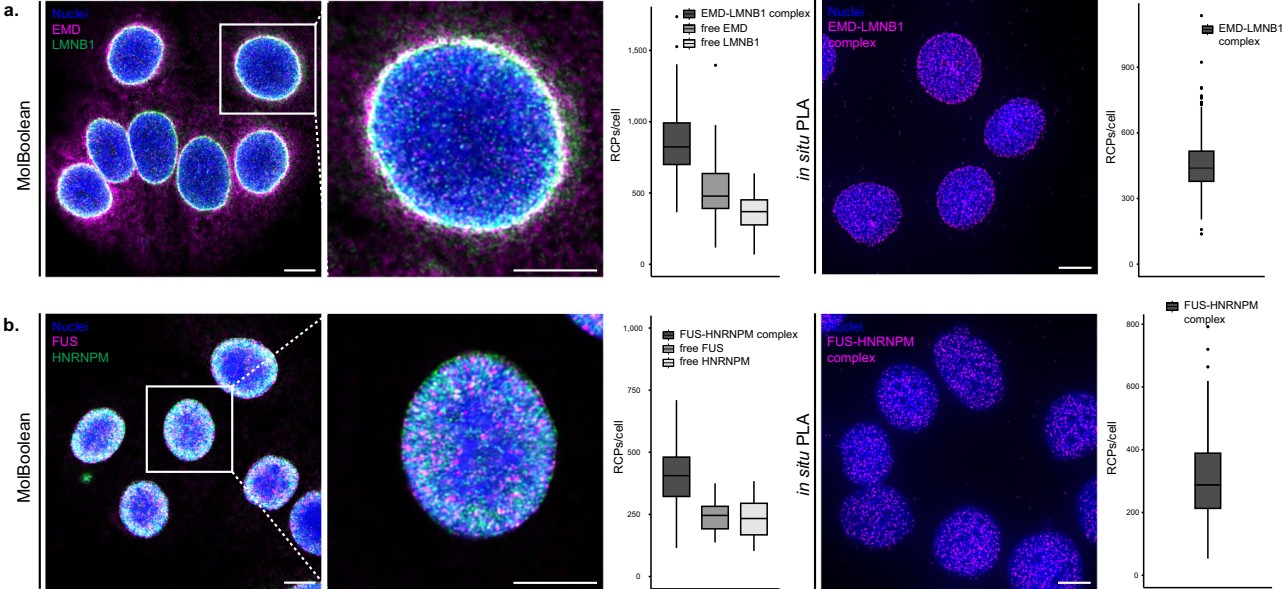

**Fig. 6 | MolBoolean and in situ PLA staining of nuclear proteins in MCF7 cells. a** EMD and LMNB1 co-stain. MolBoolean signals are shown for EMD (magenta), LMNB1 (green), EMD-LMNB1 complex (white) and nuclei (blue). In situ PLA signals for EMD-LMNB1 complex are shown in magenta and nuclei in blue. **b** FUS and HNRNPM co-stain. MolBoolean signals are shown for FUS in magenta, HNRNPM (green), FUS-HNRNPM complex (white) and nuclei (blue). In situ PLA signals for FUS-HNRNPM complex are shown in magenta and nuclei in blue. White frames

depict an area shown in enlarged view in the following panel. Scale bars = 10 µm. Quantification of protein complexes and free proteins (MolBoolean) or protein complexes only (in situ PLA) shown as number of RCPs per cell. Data pooled from three independent experiments. Box plots show median, Q1 to Q3 range, lower and upper whiskers at maximum 1.5 times the interquartile range. Source data are provided as a Source Data file.

in the nucleus with the exception of the nucleoli[25], whereas HNRNPM is a pre-mRNA-binding protein involved in splicing and found in the nucleoplasm according to the Human Protein Atlas[16] (https://www.proteinatlas.org/ENSG00000099783-HNRNPM/cell). We performed MolBoolean and found a high level of colocalization in the nucleus, but also relatively high levels of free FUS and HNRNPM (Fig. 6b). Comparable level of protein complexes (Fig. 6b, in situ PLA panel) and subcellular localization (Supplementary Fig. 4b, IF panel) were observed with classical validation methods.

**Dynamic changes in protein states under varying conditions**
After demonstrating that MolBoolean efficiently stains abundant proteins, we wanted to further explore how sensitive and specific the method is for detection of decreased amounts of one interaction partner. We performed a stain for PDIA3 (also known as ERp57) and calreticulin (CALR) in HaCaT cells (Fig. 7a and Supplementary Fig. 5a). PDIA3 is a protein primarily located in the endoplasmic reticulum (ER), where it participates in the folding of newly synthesized glycoproteins together with Calnexin and Calreticulin[26,27]. It has also been detected in the cytoplasm, cell membrane[27,28], and nucleus[29]. In addition to being localized to the ER, the lectin chaperone Calreticulin, too, has been found to localize in the cytosol[30], in association with the vitamin D

receptor, and further plays a role in nuclear export[31,32]. As determined by Western blot, we achieved 92.5% silencing of PDIA3 in HaCaT cells via siRNA-treatment (Fig. 7a, Western blot membrane and quantification below) and compared MolBoolean in the siRNA-treated cells ("PDIA3 knock-down" condition) to mock-transfected ones ("control" condition). Significant decrease of free PDIA3 was detected in the PDIA3 knock-down cells compared to the control (Fig. 7a, MolBoolean quantification), as well as significant downregulation of Calreticulin in accordance with literature[33], and a dramatic drop in complex formation. In contrast, the level of dual signal remained high in cells with normal expression of PDIA3. In situ PLA confirmed statistically significant decrease in interactions upon silencing (Fig. 7a, in situ PLA panel). IF data confirmed the subcellular protein distribution and the silencing of PDIA3 (Supplementary Fig. 5a).

While in the case of PDIA3 silencing we showed the detection of proteins with varying abundance, and with the TGF-β1 treatment of HaCaT cells, we predominantly observed a decrease of complex formation and re-localization of free signal, we further tested MolBoolean's performance with an inducible interaction that is known to increase after ligand stimulation. We focused on platelet-derived growth factor receptor β (PDGFR-β), a receptor tyrosine kinase activated by ligands such as PDGF-BB[34] (Fig. 7b and

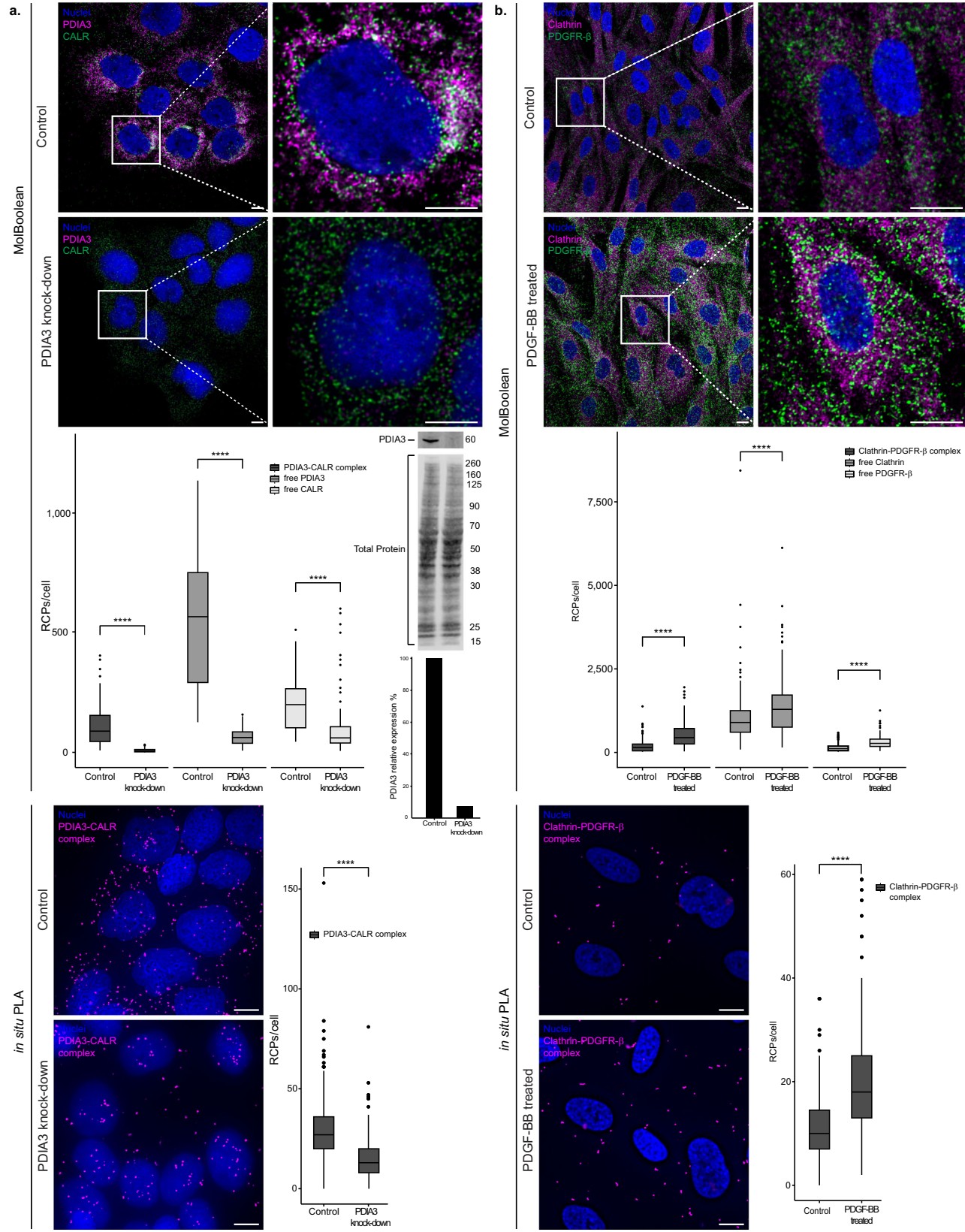

Supplementary Fig. 5b). Upon ligand-induced activation, the membrane-bound PDGFR-β is mostly internalized via Clathrin-coated pits[34,35], which has an important function in downstream signaling in the early endosomes[36]. It has been previously demonstrated with in situ PLA that upon PDGF-BB stimulation PDGFR-β in fibroblasts shows increased colocalization with Clathrin[37]. We therefore treated

BJ h-TERT cells with PDGF-BB for 0 min ("control") and 15 min ("PDGF-BB treated") accordingly, and applied MolBoolean to quantify and compare the amounts of free Clathrin and free PDGFR-β, as well as the amount of dual signal under both conditions. We also verified the latter with in situ PLA (Fig. 7b) and IF (Supplementary Fig. 5b). Clathrin–PDGFR-β colocalization increased significantly

**Fig. 7 | MolBoolean staining in dynamic conditions. a** PDIA3 and CALR co-stain in untreated HaCaT cells ("control", top), and after 72 h treatment with siPDIA3 ("PDIA3 knock-down", bottom). Membrane represents Western blot, and Western blot quantification of silencing efficiency (92.5% knockdown, based on normalization against total protein stain) is shown in the bar chart below. ($p = 2.56e-32$; $p = 2.81e-35$; $p = 1.94e-15$ for PDIA3-CALR complex, free PDIA3 and free CALR respectively (MolBoolean); $p = 2.19e-38$ (in situ PLA)). MolBoolean signals are shown for PDIA3 (magenta), CALR (green), PDIA3-CALR complex (white) and nuclei (blue). In situ PLA signals for PDIA3-CALR complex are shown in magenta and nuclei in blue. **b** Clathrin and PDGFR-β co-stain in BJ-hTERT cells, in the absence ("control", top) or presence ("PDGF-BB treated", bottom) of PDGF-BB. ($p = 6.76e-20$; $p = 5.88e-05$; $p = 1.36e-17$ for Clathrin-PDGFR-β complex, free Clathrin and free PDGFR-β respectively (MolBoolean); $p = 3.85e-22$ (in situ PLA)). MolBoolean signals are shown for Clathrin (magenta), PDGFR-β (green), Clathrin-PDGFR-β complex (white) and nuclei (blue). In situ PLA signals for Clathrin-PDGFR-β complex are shown in magenta and nuclei in blue. White frames depict an area shown in enlarged view in the following panel. Scale bars = 10 μm. Quantification of protein complexes and free proteins (MolBoolean) or protein complexes only (in situ PLA) shown as number of RCPs per cell. $n_{control} = 103$, $n_{knock-down} = 104$ cells (**a**). $n_{control} = 150$, $n_{treated} = 140$ cells (**b**). Data pooled from three independent experiments. Two-sided Wilcoxon rank sum test was used to analyze statistical variance. Box plots show median, Q1 to Q3 range, lower and upper whiskers at maximum 1.5 times the interquartile range. Outliers shown as solid circles. ****$p < 0.0001$. Source data are provided as a Source Data file.

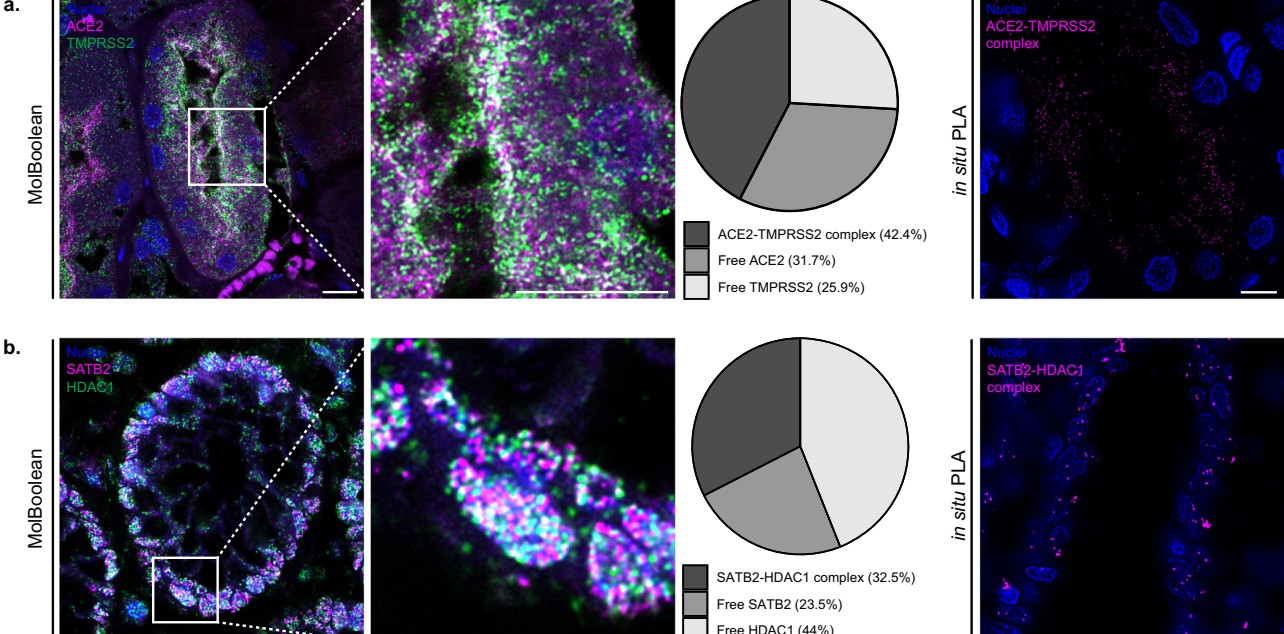

**Fig. 8 | MolBoolean and in situ PLA staining and quantification in FFPE tissue sections. a** ACE2 and TMPRSS2 co-stain in kidney. MolBoolean signals are shown for ACE2 (magenta), TMPRSS2 (green), ACE2-TMPRSS2 complex (white) and nuclei (blue). In situ PLA signals for ACE2-TMPRSS2 complex are shown in magenta and nuclei in blue. **b** SATB2 and HDAC1 co-stain in colon. MolBoolean signals are shown for SATB2 (magenta), HDAC1 (green), SATB2-HDAC1 complex (white) and nuclei (blue). In situ PLA signals for SATB2-HDAC1 complex are shown in magenta and nuclei in blue. White frames depict an area shown in enlarged view in the following panel. Scale bars = 10 μm. MolBoolean quantification shown in percentage of RCPs in each category (free protein A, free protein B and AB complex) per frame. Data collected from three independent experiments. Source data are provided as a Source Data file.

upon stimulation as detected by both MolBoolean and in situ PLA (Fig. 7b, quantifications).

## MolBoolean analysis of proteins in tissue sections

FFPE tissue sections are routinely used in histopathological analyses in research and in the clinic. In order to validate that our method can not only be used successfully in cells, but also in tissue applications, we stained kidney tissue against ACE2 and its interaction partner TMPRSS2 (Fig. 8a and Supplementary Fig. 6a). ACE2 is an important counter-regulator of the renin-angiotensin system with a role in vascular homeostasis, and an entry point of the SARS-CoV-2 virus causing the Coronavirus disease 2019 (Covid-19)[38,39]. Our analysis demonstrated its characteristic membranous expression in the proximal renal tubule cells[40] and strong colocalization with TMPRSS2 (Fig. 8a), a serine-protease that, among other functions, facilitates SARS-CoV-2 viral uptake in the cell[41,42].

Furthermore, we applied MolBoolean to detect SATB2 and HDAC1 in colon tissue (Fig. 8b and Supplementary Fig. 6b). SATB2 is a nuclear DNA-binding protein[43] which participates in chromatin remodeling by recruiting, among others, HDACs to promotors and enhancers. HDAC1

is a histone deacetylase and a prognostic marker for colorectal cancer involved in epigenetic regulation via transcriptional repression[44,45]. SATB2 is known to recruit HDAC1 to DNA[46–48], and in agreement with that we demonstrated fairly high levels of colocalization between the two proteins in the nuclei of glandular cells in the colon mucosa, accompanied by high levels of free protein both for HDAC1 and for SATB2 (Fig. 8b, pie chart).

For additional FFPE staining examples see Supplementary Fig. 6c, d, and Supplementary Notes.

## Discussion

Taken together, our results demonstrate that the MolBoolean method is versatile and works reliably in fixed cells and tissues in order to sensitively and selectively visualize both free and interacting proteins at the same time. It is efficient in discriminating single from dual signals in a wide range of organelles such as the cell membrane, ER, Golgi complex, endosomes, mitochondria, etc (e.g., Figs. 2, 4, 7, Supplementary Fig. 3c,). It even works reliably in crowded and less accessible compartments of the cell like the nucleus, as shown in the EMD-LMNB1 and FUS-HNRNPM experiments (Fig. 6). Like other approaches for

determining proximity between proteins, such as in situ PLA and FRET, MolBoolean provides information on whether the proximity probes, be it primary or secondary antibodies, have bound their targets within the distance that would allow for colocalization readout. Thus, the detection of signal with any of these proximity-based approaches should only be considered indirect proof, and cannot be used as indisputable evidence of physical interaction between the two targeted proteins. This is an important caveat, although in many cases, proximity is indeed indicative of two proteins forming a complex. The distance threshold that allows for the formation of a dual-colored RCP in MolBoolean is determined by the size of the affinity reagents used, as well as the length of the oligonucleotides. Therefore, using primary *versus* secondary antibodies as probes also affects that. In the current secondary antibody-based design this theoretical distance is similar to what has been reported for in situ PLA.

Due to the discrete, dot-like nature of RCPs, signal intensity in MolBoolean is not only amplified compared to regular immunostaining techniques, but it also allows for quantification of the number of RCPs, normalization whenever reasonable, and comparison between different conditions. This was clearly demonstrated in our TGF-β1 and PDGF-BB treatments (Figs. 5b and 7b, respectively), and in the PDIA3-silencing experiment (Fig. 7a). TGF-β1 is a well-known inducer of epithelial-to-mesenchymal transition (EMT) that leads to the acquisition of mesenchymal characteristics, increased motility and invasiveness of induced cells[20,49,50]. In culture, a reduction in the local density of HaCaT cells and rearrangement of cell adhesion structures, accompanied by decreased expression of E-cadherin at the membrane and increased cytoplasmic localization have been described in response to TGF-β1 stimulation[20]. In line with literature, our prolonged TGF-β1 treatment of HaCaT cells leads to observable modifications in cell morphology and increased migration (i.e. cells lose contact and spread out over a larger surface area), as well as redistribution of free and interacting proteins. This highlights the importance of being able to simultaneously monitor free and complex-bound states. Simply performing in situ PLA in this case would not be informative, as it would be easy to deduce that there is an increase in E-cadherin–β-catenin interactions post-treatment (Fig. 5b). This misleading observation would be true in absolute numbers, but not in relation to the total number of the two proteins recorded in each enlarged cell. IF, on the other hand, shows the morphological changes, but cannot be used for quantification of the increased amounts of free cytoplasmic E-cadherin after the addition of TGF-β1 (Supplementary Fig. 5b). The TGF-β1 and the PDGF-BB treatment regimens both are a demonstration of the ability of MolBoolean to sensitively capture the dynamic changes of protein complex formation under different conditions.

An important characteristic of the MolBoolean method is its ability to discriminate between RCPs produced by actual colocalization, and closely positioned RCPs generated by two free proteins in conditions of high abundance. This was demonstrated by using AGS cells transfected with either (WT) E-cadherin, or a mutant form with decreased ability to bind the interaction partner β-catenin (Fig. 5a). Both conditions produce an abundance of the two proteins, but complex formation is recorded to significantly higher levels where WT E-cadherin is expressed. In another assay where we compared a cell line not expressing one interaction partner (U2OS does not express E-cadherin, but does express β-catenin) to a cell line that expresses both (MCF7), we showed that MolBoolean detects free and complex-bound E-cadherin only in the MCF7 cells, whereas free β-catenin was observed in both (Fig. 3a). These results showcase the MolBoolean sensitivity and specificity.

The possibility to discern between free proteins and interacting partners in a single assay is further advantageous in that the parallelization allows for detection on just one tissue slide, thereby saving time and materials (e.g., Figs. 5c, 8, Supplementary Figs. 3c, 6). This can be especially valuable in clinical use, where availability of consecutive

tissue sections for diagnostic staining might be limited, but also in research laboratories, where understanding the dynamics of protein complex formation in a group of cells or one cell at a time might be of interest.

Like all immunostaining methods, MolBoolean is dependent on the quality of the antibodies used. Rigorous validation of antibody specificity is required to ensure that the antibodies actually target their intended proteins[51]. However, an advantage of MolBoolean is that it offers specific staining of single proteins by means of dual recognition of two different epitopes within the same target, which also allows identification of cross-reactivity (Fig. 2). Our M-β-catenin–R-β-catenin assay demonstrated the expected pattern of staining in MCF7 cells and showcased the ability of MolBoolean to detect many more protein molecules per cell compared to in situ PLA. At the same time, it also highlighted once again that all immuno-based methods are highly reliant on antibody affinity, and that less-than-ideal conditions (which are almost inevitably the case in reality) lead to some off-target staining in the form of single-colored signals. The number of reported free proteins *versus* proteins in complex is a relative measurement, where the ratio is dependent on antibody binding, efficiency of hybridization, and subsequent enzymatic steps. The concentrations of the antibodies used need to be high enough to ensure that the majority of epitopes are bound. Detection of interacting proteins depends on antibodies binding both targets, like for in situ PLA and antibody-based FRET. If, for example, 80% of all available epitopes are bound by an antibody, then 64% (i.e., $80\% \times 80\%$) of the protein complexes will be bound by both antibodies. Therefore, one should aim to saturate as many epitopes as possible in order not to disadvantage dual signal detection. To decrease off-target effects, primary antibody conjugates may be used as probes (Supplementary Fig. 3c), since this eliminates the background from any unspecific binding of the secondary antibodies. In addition to antibody binding, MolBoolean relies on several enzymatic steps. For the recording of dual signal, it is necessary that the information receiver circle is nicked in two places and two tag oligonucleotides are successfully incorporated. Although the efficiency of the enzymatic steps is very high, as demonstrated by in solution tests (Supplementary Fig. 1), any reduction will favor the generation of single-colored RCPs.

Compared to in situ PLA where crowding of highly expressed proteins might generate false positive detection, MolBoolean offers additional information in terms of identifying non-interacting fractions of each protein, which is especially useful for stains such as the one against E-cadherin and β-catenin, or in the assays where we stained against abundant proteins located in different compartments (e.g., MT-CO1 and GM130 (Supplementary Fig. 2c, d), or E-cadherin and Lamin A/C (Fig. 3b)). Taken together with the information about the expected background from antibody cross-reactivity (which can be deduced from the quantification of omitting controls), MolBoolean thus allows the conclusion that the protein pairs in the examples above do not form complexes. Still, for situations where the staining results in extremely abundant signals for the proteins of interest, the detection of interactions becomes less reliable, as image analysis will report some adjacent RCPs as one dual-stained object. Further improvements in image analysis, such as 3D analysis, will likely reduce or eliminate this issue. In addition, to refine the results, there is also a possibility to use FRET[52] between the different fluorophores on detection oligonucleotides A and B to determine if they are situated within the same RCP or not.

In conclusion, MolBoolean provides opportunities for studying biological processes, has applications in diagnostics, and decreases the risk of false positive signals compared to in situ PLA.

## Methods

### Ethical statement

This study includes samples of anonymized formalin-fixed paraffin-embedded (FFPE) human tissue samples from ovarian carcinomas that

were collected under local ethical guidelines, with informed consent (as stipulated by the Declaration of Helsinki) and approved by the Ethical Committee from Centro Hospital de São João (CHSJ) (Ref.86/2017), with permission to publish data generated.

## Cell culture and tissue sections

All cell lines were cultured in standard conditions (37 °C, 5 % v/v $CO_2$) in a humidified incubator and were grown in complete medium (i.e. medium supplemented with 10% FBS), unless in starvation and/or stimulation conditions, in which case the medium was supplemented with either a very low percentage of FBS, or no FBS at all (referred to as starvation medium).

HaCaT and BJ-hTERT cells were cultured in Dulbecco's Modified Eagle's Medium (DMEM) supplemented with GlutaMAX™-I and 10% (v/v) Fetal Bovine Serum (FBS) (all from Thermo Fischer Scientific). MCF7 cells (ECCAC 86012803) were cultured in Minimum Essential Medium Eagle (EMEM) with additives: 2 mM Ala-Gln, 1% Non-Essential Amino Acids (NEAA), and 10 % (v/v) FBS, all from Sigma-Aldrich. AGS cell stable clones (E-cadherin WT and V832M, a kind gift from Raquel Seruca, University of Porto) were cultured in Roswell Park Memorial Institute (RPMI) 1640 medium, 10% (v/v) FBS and 1% penicillin-streptomycin (all from Sigma-Aldrich), and supplemented for selection with 10 ng/μL blasticidine (Gibco), renewed every 3-4 days. U2OS cells were purchased from ECACC (cat no: 92022711). They were cultured in McCoy's 5a medium supplemented with 10% (v/v) FBS and 2 mM Ala-Gln (all from Sigma-Aldrich).

For the PDIA3 silencing assay, Silencer Select siRNA (Thermo-Fisher) was used to transfect HaCaT cells, seeded at a density of 150.000 cells/ml. Either siPDIA3 (ThermoFisher, s6228) ("PDIA3 knock-down" condition), or Silencer™ Select Negative Control No.1 siRNA ("control" condition) were used in a final concentration of 100 nM. Briefly, either type of siRNA was used to transfect HaCaT cells for 72 h using siLentFect™ Lipid Reagent for RNAi (BioRad) according to BioRad's instructions. Afterwards, either whole-cell lysate was prepared with LDS sample buffer (Thermo-Fischer, NP0007) for testing of silencing efficiency via Western blot, or cells were fixed in 3.7% PFA on ice for IF, in situ PLA and Mol-Boolean staining. For an example of presentation of full scan blots, see Supplementary Information.

For the disruption of cell-to-cell adhesion assay (Fig. 5a), HaCaT-cells were either stimulated with 2 ng/mL TGF-β1 in DMEM starvation medium (0% FBS) for 48 h with fresh medium replacement after 24 h ("treated" condition), or grown in DMEM starvation medium (0.5% FBS) ("control" condition).

For the Clathrin−PDGFR-β assay, as described in ref. 37, BJ-hTERT cells were starved overnight in DMEM starvation medium (0.2% FBS), and then either stimulated with 20 ng/mL PDGF-BB in for 15 min ("treated" condition), or left untreated in the same medium ("control" condition).

Anonymized formalin-fixed paraffin-embedded (FFPE) human tissue samples from ovarian carcinomas were collected under local ethical guidelines, with informed consent (as stipulated by the Declaration of Helsinki) and approved by the Ethical Committee from Centro Hospital de São João (CHSJ) (Ref.86/2017). Anonymized FFPE tissue blocks for all other tissues were purchased from a commercial biobank (Asteand Biosciences/BioIVT) and used in agreement with the terms and conditions of sale. Glass slides with the tissue sections were deparaffinized by 3 × 3 min washes in xylene (Sigma-Aldrich), followed by one wash in 100% xylene and 99.9% ethanol in 1:1 ratio for 3 min, and 2 × 3 min washes with 99.9% ethanol, as well as 1 × 3 min 96% ethanol, 1 × 3 min 70% ethanol and 1 × 3 min 50% ethanol. The tissue slides were rinsed in deionized water and antigen retrieval was performed with Tris-EDTA pH 9 (DAKO) in a pressure cooker at pressure 2 atm at 95 °C for 40 min.

## In situ PLA

All in situ PLA experiments were performed with the Duolink® In Situ Red Starter Kit Mouse/Rabbit (Sigma Aldrich) according to the manufacturer's instructions. In brief, PFA-fixed cells on 8-well Nunc™ Lab-Tek™ II CC2™ Chamber Slides (Sigma-Aldrich) or deparaffinized FFPE sections that had undergone antigen retrieval (for details see Cell culture and tissue sections) were encircled with ImmEdge Hydrophobic Barrier PAP Pen (Vector Laboratories) to ensure the reaction mixes in the next steps will cover the cells/tissue. The cells were then permeabilized in 1× TBS (Thermo Fisher Scientific) 0.2% v/v Triton X-100 (Sigma-Aldrich) for 10 min, followed by 2 min wash with 1× TBS. Blocking was done with Odyssey blocking buffer (LiCor) for 1 h in a humidified chamber, and afterwards the cells were incubated with either a mixture of two primary antibodies against the respective proteins of interest, raised in different hosts (mouse or rabbit), or with either one of these antibodies (to serve as omitting controls). Primary antibodies were incubated overnight at 4 °C, followed by 3 × 5 min wash in 1× TBS, and an incubation with a mix of the Duolink® In Situ PLA® Probe Anti-Rabbit PLUS, Affinity purified Donkey anti-Rabbit IgG (H + L) and Duolink® In Situ PLA® Probe Anti-Mouse MINUS, Affinity purified Donkey anti-Mouse IgG (H + L) (all from Sigma-Aldrich) in the concentrations recommended by Sigma-Aldrich. Next followed hybridization and ligation with the Duolink® Ligation mix, and finally, RCA and signal detection were performed using the Duolink® In Situ Detection Reagent Red. Nuclei were labeled with Hoechst33342 (1:250, Thermo Fisher Scientific). The slides were then mounted with Slow-Fade Gold antifade reagent (Thermo Fisher Scientific), and images were acquired with Zeiss AxioImager M2 with a Zeiss Plan-Apochromat 63x NA 1.4 oil objective and deconvolved with Huygens Essential (Scientific Volume Imaging, the Netherlands, http://svi.nl) using the Deconvolution Wizard option. Quantification and colocalization analyses were performed with the CellProfiler[50] software on the deconvolved but otherwise unaltered images.

## IF staining

PFA-fixed cells or FFPE tissues (for preparation, see Cell culture and tissues) were stained using standard immunofluorescence techniques. Permeabilization was performed as for in situ PLA, and subsequently both cells and tissues were blocked with Odyssey blocking buffer (LiCor) for 1 h. After blocking, the primary antibodies of interest were diluted in blocking buffer and applied to the slides overnight at 4 °C in a humidified chamber, and afterwards washed in 1× TBS for 3 × 5 min. Next, fluorophore-labeled secondary antibodies and Hoechst33342 (1:250, Thermo Fisher Scientific) were added for 1 h at 37 °C, followed by 1× TBS-Tween-20 wash and mounting with SlowFade Gold antifade reagent (Thermo Fisher Scientific). Images were acquired with Zeiss AxioImager M2 with a Zeiss Plan-Apochromat 63x NA 1.4 oil objective and then deconvolved with Huygens Essential (Scientific Volume Imaging, the Netherlands, http://svi.nl) using the Deconvolution Wizard option.

## MolBoolean sequence design

All MolBoolean oligonucleotide sequences (Table 1) were designed by hand and tested in Nupack[53] (nupack.org) for the formation of secondary structures and hybridization at different concentrations, temperatures and salinity.

## Padlock probes

We designed padlock probes (see Table 1) as previously described[17]. Each probe is 99 nt long, out of which 25 nt in the 5′ end and 24 nt in the 3′ end are complementary to the MolBoolean arms. Hybridization to the arm will bring the 5′- and 3′ end together and act as a template for ligation. The 5′- and 3′ ends of the padlock probes are joined by a 50 nt linker.

**Table 1 | DNA design for MolBoolean**

| Name | Modification | Sequence (5' → 3') |
|---|---|---|
| Circle part 1 | 5' phosphate | TTTATCTATATCTGCCACGCTACTTACGTCTCTCGTCTGATGCTCCACCTCATATATA AATTGTGTCCACTCGTCTCACTGCTCAACTACCTACCTCAGGAGAAACCTTTACTT |
| Circle part 2 | 5' phosphate | CGAGGTGCTTTTAGCACCTCGAAGTAAAGCTATCCACTGTCACCAACTACTA GATAAACGTCACACTTTTCGTGTGACG |
| Arm A | 5' aldehyde or oYo-link | AAAAAAAAACTCCTGAGGTAGGTAGTTGAGCAGCATCCGCACTTATAGC TGCAGTGAGACGAGTGGACAC |
| Arm B | 5' aldehyde or oYo-link | AAAAAAAAATGAGGTGGAGCATCAGACGGTAATTAACCCGCCCCGTACGAGA GACGTAAGTAGCGTGGCA |
| Tag A | 5' phosphate | CTGCAGCTATAAGTGCGGATG |
| Tag B | 5' phosphate | CGTACGGGGCGGGTTAATTAC |
| Detection oligo A | 5' Texas Red / Atto565 2'-MeO-U | CTGCAGCTATAAGTGCGGATGUUU |
| Detection oligo B | 5' Atto647N 2'-MeO-U | CGTACGGGGCGGGTTAATTACUUU |
| Padlock A | 5' phosphate | AGTGCGGATGCTGCTCAACTACCTACACCTCGAAGTAAAGCTATCCACTGTCACCAACTACT AGATAAACGTCACTCCACTCGTCTCACTGCAGCTATA |
| Padlock B | 5' phosphate | GGTTAATTACCGTCTGATGCTCCACACCTCGAAGTAAAGCTATCCACTGTCACCAACT ACTAGATAAACGTCACCTACTTACGTCTCTCGTACGGGGCG |

## Circle ligation

The circle parts 1 and 2 (Table 1) were ligated by using 0.02 U/µl T4 ligase (Thermo Fisher Scientific) in T4 DNA ligase buffer (Thermo Fisher Scientific) for 2 h at room temperature. Non-ligated oligonucleotides were removed via digestion with a mixture of 2 U/µl exonuclease I, 0.5 U/µl lambda exonuclease, and 0.5 U/µl T7 exonuclease (all from New England Biolabs) in 1× exonuclease I reaction buffer (New England Biolabs) at 37 °C overnight, followed by heat inactivation of the enzymes at 80 °C for 30 min and subsequent validation by gel electrophoresis.

## NHS-ester conjugation of MolBoolean proximity probes

The antibody components of the probes were concentrated to a minimum of 2 µg/µl using Amicon Ultra-15 Centrifugal Filter Units (Sigma-Aldrich). Succinimidyl 6-hydrazinonicotinate acetone hydrazine (SANH) Crosslinker (Solulink) was added at a 25-molar excess to the antibodies, and incubated under gentle agitation for 2 h at room temperature, protected from light. Each conjugation reaction underwent a buffer exchange to 100 mM NaH$_2$PO$_4$ (Sigma-Aldrich), 150 mM NaCl (Sigma-Aldrich), pH 6, with the use of Zeba Spin Desalting Columns, 7 K MWCO (Life Technologies), according to manufacturer's instructions. Activated antibodies were incubated in 100 mM NaH$_2$PO$_4$ 150 mM NaCl, pH 6 with a 3-molar excess of aldehyde-modified arm A (for anti-mouse IgG) or B (for anti-rabbit IgG) and 10 mM aniline (Sigma-Aldrich) as a catalyst. The reactions were incubated protected from light and with gentle agitation for 2.5 h in room temperature, before a buffer exchange to 1× PBS (Thermo Fischer Scientific), followed by size-exclusion purification.

## oYo-Link conjugation of primary MolBoolean proximity probes

For preparation of MolBoolean probes using direct conjugation of primary antibodies to arm oligonucleotides A and B respectively, 100 µg of anti-mouse E-cadherin (AMAb90862, Atlas Antibodies) and 100 µg of anti-rabbit β-catenin (AMAb91209, Atlas Antibodies) primary antibodies (Table 2) in PBS formulation were used. The next steps were performed according to the recommended protocol for oYo-Link™ conjugation (AlphaThera). In brief, 5'-oYo-Link-modified arms A and B (Table 1) were ordered from AlphaThera via the oYo-Link Oligo Custom option, and each arm was resuspended in 100 µL of nuclease-free water in accordance with the manufacturer's instructions. Each antibody was then mixed well with the corresponding arm (1 µg of antibody per 1 µL of oYo-modified arm), centrifuged, and subjected to Light-Activated Site-Specific Conjugation (LASIC) under 365 nm black light for 120 min on ice, in order to achieve covalent binding of oligo to antibody. Size-exclusion purification followed.

## Size-exclusion purification

The conjugated probes were purified from unconjugated antibody and oligonucleotide by ÄKTA Pure chromatography (GE Healthcare) using a Superdex 200 10/300 GL column (GE Healthcare). Successful purification was confirmed by separating the conjugates on a Novex TBU 10% gel (Life Technologies) at 150 V for 60 min in a water bath preheated to 50 °C. DNA was visualized using SYBR Gold Nucleic Acid Gel Stain (Life Technologies), and protein was visualized using Coomassie brilliant blue stain (Bio-Rad). The gel was imaged on Odyssey Fc with the Image Studio Lite v5.2.5 software (LI-COR Biosciences).

## In solution specificity tests

All reagents were diluted to concentrations corresponding to those used in situ in proportion to the reaction volume. Unhybridized MolBoolean oligonucleotides were used as size references (Supplementary Fig. 1, wells 1 through 5 on the gel represent the ligated circle, arm A, arm B, tag A, and tag B respectively). To demonstrate the specificity of the nickase, we prepared a 2x Digestion Master mix (0.25 U/mL Nt.BsmAI in water and 2x NEBuffer CutSmart (both from New England Biolabs)) and mixed it with circle in a final concentration of 0.1 µM. A quarter of this reaction mix was set aside, and the rest was divided into three equal parts to which we added as follows: arm A at a final concentration of 0.2 µM; or arm B at a final concentration of 0.2 µM; or arms A and B at a final concentration of 0.2 µM each. The resulting four digestion reactions were incubated at 37 °C for 1 h, and then at 65 °C for 20 min in order to heat-inactivate Nt.BsmAI according to the manufacturer's instructions (shown in Supplementary Fig. 1, wells 6 through 9 respectively). Next, we proceeded by preparing five ligation reaction mixes on the basis of the digestion mixes from the previous step. For the reaction mix in Supplementary Fig. 1, well 10, we added tag A at a final concentration of 1 µM to the mixture of nicked circle and arm A. In the same way, for the reaction mix shown in well 11, we added tag B to the nicked circle and arm B mix. For the mix in lane 12, we combined both tags A and B and added them to the mixture of nicked circle and both arms prepared at the previous step. In addition, for the mixture in well 13, we added tag B to the mix of nicked circle and arm A, whereas for the mixture in well 14, we added tag A to the mix of nicked circle and arm B. These five reactions (Supplementary Fig. 1, wells 10–14) all had 1x T4 Ligation buffer and 0.05 U/µL of T4 Ligase (both from ThermoFisher) added after hybridization between tags and arms was allowed for 30 min at 37 °C first. After enzyme addition, the five samples were incubated for another 30 min at 37 °C to allow for ligation. Next, the ligase in the samples was heat-inactivated at 80 °C for 20 min as per the enzyme's manufacturer's recommendations. All samples were loaded on a denaturing Novex TBU 10% gel (Life

**Table 2 | Antibodies used for MolBoolean experiments**

| Antibody | Dilution/working concentration | Company, serial no, lot number |
|---|---|---|
| Mouse anti-β-catenin Clone CL3689 | 2,5 µg/mL | Atlas Antibodies, AMAb91209, 03052 |
| Rabbit anti-β-catenin | 2,5 µg/mL | Atlas Antibodies, HPA029159, B115015 |
| Mouse anti E-cadherin Clone CL1170 | 5 µg/mL | Atlas Antibodies, AMAb90862, 03700 |
| Mouse anti-E-cadherin Clone 36 | 2,5 µg/mL | BD Transduction Laboratories, #610182, 2104735 |
| Rabbit anti-Lamin A/C | 1:100 | Cell Signaling, #2032, 6 |
| Rabbit anti-β-catenin Clone D10A8 | 1:100 | Cell Signaling, #8480, 5 |
| Mouse anti-EMD Clone CL0203 | 5 µg/mL | Atlas Antibodies, AMAb90562, 02602 |
| Rabbit anti-LMNB1 | 2 µg/mL | Atlas Antibodies, HPA050524, R60423 |
| Mouse anti-FUS CloneCL0190 | 5 µg/mL | Atlas Antibodies, AMAb90549, 03186/03693 |
| Rabbit anti-HNRNPM | 2 µg/mL | Atlas Antibodies, HPA024344, A96210 |
| Mouse anti-PDIA3 Clone MapERP57 | 10 µg/mL | BioRad, VMA00477,160801 |
| Rabbit anti-Calreticulin CloneD3E6 | 1:100 | Cell Signaling, #12238, 5 |
| Mouse anti-Clathrin Clone X22 | 1:500 | Abcam, ab2731, GR3412763-1 |
| Rabbit anti-PDGFRβ Clone 28E1 | 1:200 | Cell Signaling, #3169, 13 |
| Mouse anti-ACE2 Clone CL4013 | 5 µg/mL | Atlas Antibodies, AMAb91259, 03083 |
| Rabbit anti-TMPRSS2 | 2 µg/mL | Atlas Antibodies, HPA035787, 18833 |
| Mouse anti-SATB2 Clone CL0323 | 5 µg/mL | Atlas Antibodies, AMAb90682, 03684 |
| Rabbit anti-HDAC1 | 2 µg/mL | Atlas Antibodies, HPA029693, A96201 |
| Mouse anti-GM130 Clone 35 | 5 µg/ml | BD Transduction Laboratories, #610822, 6217559 |
| Rabbit anti-COX1/ MT-CO1 | 1:80 | Cell Signaling, #62101, 1 |
| Mouse FLEX anti-human CA 125 Clone M11 | Undiluted | Agilent, IR70161-2, 20081575 |
| Rabbit anti-Mesothelin Clone SP74 | 1:50 | Thermo Fisher Scientific, MA5-16378, GR3224762-47 |
| Rabbit β-Actin | 1:1000 | Abcam, ab8227, GR3195358-1 |

Technologies) after boiling in 50% urea for 5 min at 95 °C, and the gel was run at 130 V for 35 min in a 65 °C water bath. DNA was visualized using SYBR Gold Nucleic Acid Gel Stain (Life Technologies). The gel was imaged on Odyssey Fc with the Image Studio Lite v5.2.5 software (LI-COR Biosciences).

**MolBoolean experimental procedure**

Cells were seeded in the desired density on 8-well Nunc™ Lab-Tek™ II CC2™ Chamber Slides (Sigma-Aldrich) and treated according to the experimental condition. Fixation was performed with ice-cold 3.7% PFA (Sigma-Aldrich) for 15 min on ice. The chamber slides were dried and stored at −20 °C until use or used fresh. The wells were removed from the slides and subsequently lined with ImmEdge Hydrophobic Barrier PAP Pen (Vector Laboratories). The cells were permeabilized with 1× TBS (Thermo Fisher Scientific) 0.2% v/v Triton X-100 (Sigma-Aldrich) for 10 min, followed by 2 min wash with 1× TBS. Blocking was done either with Odyssey blocking buffer (LiCor), or homemade blocking buffer (2% BSA w/v (Jackson Immunoresearch) in 1× TBS 0.1% Tween (Sigma-Aldrich) 0.02% sodium azide (Sigma-Aldrich)), either one supplemented with 2.5 mg/mL salmon sperm DNA (Thermo Fisher Scientific) for 1 h at 37 °C in a humidified chamber. The cells were then incubated with pairs of mouse and rabbit primary antibodies against the proteins of interest, diluted in either Odyssey blocking buffer, or homemade blocking buffer, and were incubated overnight at 4 °C in a humidified chamber, followed by 3 × 3 min washes in 1× TBS. The primary antibodies used are shown in Table 2. The cells were incubated with 3 µg/mL of each proximity probe (A and B), diluted in either Odyssey blocking buffer, or homemade blocking buffer for 1 h at 37 °C in a humidified chamber, followed by 1 × 3 min wash in 1× TBS 1 M NaCl (Thermo Fisher Scientific) 0.05% v/v Tween-20, followed by 2 × 3 min wash in 1× TBS 0.05% Tween-20 (TBS-T). Subsequently, the cells were incubated in 1× T4 DNA ligase buffer, supplemented with 0.25 mg/mL BSA (Sigma-Aldrich) with 0.05 µM circle for 1 h at 37 °C in a humidified chamber, followed by 3 × 3 min wash with 1× TBS-T. Afterwards a mix of 0.125 U/µl Nt.BsmAI in 1× NEBuffer CutSmart (New England Biolabs),

and 0.25 mg/mL BSA was added for 30 min at 37 °C in a humidified chamber, followed by 3 × 3 min wash with 1× TBS-T. For the hybridization of the tag oligonucleotides, the cells were incubated in 1× TBS, 0.25 mg/mL BSA, and 0.5 µM tag oligonucleotides A and B (Table 1) for 30 min at 37 °C in a humidified chamber and ligated in 1× T4 DNA ligase buffer, 0.25 mg/mL BSA, 0.05 U/µl T4 ligase for 30 min at 37 °C, followed by 1 × 3 min wash with 1× TBS 1 M NaCl 0.05% v/v Tween-20, and 1 × 3 min wash with 1× TBS-T. For the RCA, the cells were incubated in 1x phi29 polymerase buffer (Monserate), 0.25 mg/mL BSA, 1.25 mM dNTPs (Thermo Fisher Scientific), and 1 U/µl phi29 polymerase (Monserate) for 90 min at 37 °C in a humidified chamber, followed by 2 × 10 min wash with 1× TBS-T and then incubated in 1× TBS 1 M NaCl 0.05% v/v Tween-20, 0.25 mg/mL UltraPure Salmon Sperm DNA Solution, Hoechst33342 (1:250) (Thermo Fisher Scientific), 0.025 µM detection oligonucleotides A and B (Table 1) for 30 min at 37 °C in a dark humidified chamber, followed by 1 × 10 min wash with 1× TBS 1 M NaCl, 1 × 10 min wash with 1x TBS, and 1 × 5 min wash with 0.2× TBS in the dark. Slides were mounted with SlowFade Gold antifade reagent (Thermo Fisher Scientific) according to manufacturer's instructions and sealed with Menzel Gläser coverglass #1.5 (VWR). During cell and FFPE tissue section imaging, at least three images per well or FFPE tissue section were taken in a single focal plane according to the Nyquist criteria. The microscope images were acquired using either a Zeiss AxioImager M2 (Fig. 4, all IF images) or a Leica TCS SP8 X microscope (all other images) using the Zen Blue 2 or the LasX software respectively. The former was used with a 63x/1.4 oil apochromat (Zeiss) objective lens, a Hamamatsu C11440 camera and an HXP 120 V (Zeiss) light source for excitation. The latter was used with a water immersion HC PL APO 63x/1.20 NA, motCORR CS2 objective lens (Leica), and the Leica White light Laser. Images were deconvolved with Huygens Essential (Scientific Volume Imaging, the Netherlands, http://svi.nl) using the Deconvolution Wizard option. Quantification and colocalization analyses were performed with the CellProfiler[54] software on the deconvolved but otherwise unaltered images. Adjustments of brightness and contrast were then made on figure images for

visualization purposes only. Pseudo-coloring was applied to all images; Hoechst33342, Texas Red, and Atto647N are depicted in blue, magenta, and green respectively.

## Data analysis

Deconvolved split-channel images in grayscale.tif format were analyzed using the CellProfiler software version 4.1.3[54]. For Mol-Boolean analysis, a pipeline for signal quantification, with slight modifications between assays, was compiled with the following modules: *IdentifyPrimaryObjects*, *IdentifySecondaryObjects*, *EnhanceOrSuppressFeatures*, *IdentifyPrimaryObjects*, *ExpandOrShrinkObjects*, *CombineObjects*, *MaskObjects*, *MeasureObjectIntensity*, *DisplayDensityPlot*, *ClassifyObjects*, *FilterObjects*, *OverlayOutlines*, *SaveImages*, *MeasureImageAreaOccupied*, *RelateObjects*, *ConvertObjectsToImage*, *GrayToColor*, and *ExportToSpreadsheet*. First, *IdentifyPrimaryObjects* was used on the nuclear stain channel to identify nuclei based on their diameter measured in pixels and the application of two-class Otsu thresholding. Thereafter, *IdentifySecondaryObjects* was used to identify the cells, by means of expanding the nuclei by distance−N, by a certain number of pixels set by user. The images with the RCPs were filtered to remove background with a white top-hat filter through the enhance speckles feature in the *EnhanceOrSuppressFeatures* module and *IdentifyPrimaryObjects* was used in both filtered images containing RCPs, in order to identify the RCPs within a certain diameter (measured in pixels), using two-class Otsu or global manual thresholding. The identified RCPs were then shrunk to a single point in the *ExpandOrShrink* module and expanded again by distance−B to a certain number of pixels, using minimum cross-entropy thresholding, so as to better encapsulate the RCPs, using the *IdentifySecondaryObjects* module. The identified expanded RCPs were merged in the *CombineObjects* module and any signal outside the defined cells was removed through the *MaskObjects* module in order to avoid inclusion of unspecific background specks or RCPs from cells that have been excluded from the analysis. The intensity of the RCPs in both channels was measured with the *MeasureObjectIntensity* module and plotted on a density plot through the *DisplayDensityPlot* module. In the *ClassifyObjects* module, a manually determined threshold based on the density plot was used for the classification of objects, where the aim was to avoid background and include the highest intensity signal. In preparation for downstream analysis, the manual thresholds for both images were set to zero. The RCPs were then filtered by means of the *FilterObjects* module, based on their classification as protein complex or free protein and the bins of RCPs generated, were saved as images with the nuclei outlines overlaid, through the *SaveImages* and *OverlayOutlines* modules. Thereafter, the RCPs were assigned to cells with the *RelateObjects* module. In order to get a quality control of the segmentation and classification of the RCPs, the classified RCPs were shrunk again to a single point through the *ExpandOrShrinkObjects* module and converted to binary images through the *ConvertObjectsToImage* module, on which the outline of the RCPs was overlaid by means of the *OverlayOutlines* module. *GrayToColor* module was used in order to generate a quality control image, consisting of the original images with RCPs, as well as the reduced to single point RCPs, classified to bins of protein complex and free proteins. The color-coded outlines of the RCPs assigned classification were overlaid to the quality control image through the *OverlayOutlines* module and the generated images were saved through the *SaveImages* module. The data were thereafter exported to a CSV file with the *ExportToSpreadsheet* module and were used for downstream analysis in which data were binned by applying angled thresholds based on the intensities of every class of signal (analysis with description and examples is available at Github: https://github.com/pharmbio/molboolean_code[55]), and

statistics. For an example of specific settings that we used for our CellProfiler pipeline, see Supplementary Notes.

For in situ PLA analysis, a pipeline for signal quantification, with slight modifications between assays, was compiled with the following modules: *IdentifyPrimaryObjects*, *IdentifySecondaryObjects*, *EnhanceOrSuppressFeatures*, *IdentifyPrimaryObjects*, *MaskObjects*, *RelateObjects*, and *ExportToSpreadsheet*. First, *IdentifyPrimaryObjects* was used on the nuclear stain channel to identify nuclei based on their diameter measured in pixels and the application of three-class Otsu thresholding. Thereafter, *IdentifySecondaryObjects* was used to identify the cells, by means of expanding the nuclei by Distance - N, by a certain number of pixels. The images with the RCPs were filtered to remove background with a white top-hat filter through the enhance speckles feature in the *EnhanceOrSuppressFeatures* module and *IdentifyPrimaryObjects* was used in the filtered images containing RCPs, in order to identify the RCPs within a certain diameter (measured in pixels), using robust background or two-class Otsu thresholding. Thereafter, any signal outside the defined cells was removed through the *MaskObjects* module in order to avoid inclusion of unspecific background specks or RCPs from cells that have been excluded from the analysis and the RCPs were assigned to cells with the *RelateObjects* module. The data were thereafter exported to a CSV file with the *ExportToSpreadsheet* module and were used for downstream analysis and calculations (R code is available at https://github.com/pharmbio/molboolean_code[55]). For an example of specific settings we used for our CellProfiler pipeline, see Supplementary Notes.

Data were quantified as number of single- and dual-colored RCPs either recorded per individual cell (in the case of fixed cell stains), or as detected in an image frame (for tissue stains).

## Statistics and reproducibility

No statistical method was used to predetermine sample size. No data were excluded from the analyses. The experiments were not randomized. The investigators were not blinded to allocation. Statistical differences were analyzed using nonparametric tests two-sided Wilcoxon rank sum test for comparison of two groups or Kruskal–Wallis and two-sided Dunn's test with Bonferroni correction for comparison of three or more groups.

## Reporting summary

Further information on research design is available in the Nature Research Reporting Summary linked to this article.

## Data availability

All data generated or analyzed during this study are included in this published article and its Supplementary Information. Source Data are also provided with this paper, containing the data underlying the quantifications and statistical analyses; CellProfiler output files can be found in the Zenodo repository under https://zenodo.org/record/6923187#.YuJ1n3ZBwuU[56]. Source data are provided with this paper.

## Code availability

Custom code with examples on how it was used in data analysis of MolBoolean and in situ PLA is available in the Github repository under https://github.com/pharmbio/molboolean_code[55].

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

## Acknowledgements

This study has been funded by grants from the Swedish Foundation for Strategic Research (O. Söderberg), the Swedish Cancer Foundation (O. Söderberg) and the Swedish Research Council (O. Söderberg). AKle was funded via the BioImage Informatics Facility, a unit of the National Bioinformatics Infrastructure Sweden NBIS, with funding from SciLife-Lab, National Microscopy Infrastructure NMI (VR-RFI 2019-00217), and the Chan-Zuckerberg Initiative.

## Author contributions

O. Söderberg invented the MolBoolean method and supervised the work. D.R. and O. Söderberg designed the MolBoolean oligonucleotides, and D.K. designed the padlocks. D.R., D.K., and T.M. performed MolBoolean experiments, microscopy imaging and image deconvolution. D.R., D.K., M.R.S., M.L., and J.H. performed in situ PLA and IF experiments and microscopy thereof. D.K., C.S., D.R., and M.L. performed antibody conjugations for production of the MolBoolean probes. S.R., L.D., J.H., K.V., A.D., and C.K. provided materials such as cells and tissue sections, and/or performed cell treatments necessary for multiple MolBoolean experiments. J.H., A.S., and M.N. provided knowledge and ideas on interactions to be tested with MolBoolean, and how treatments need to be performed, and provided some reagents. P.J.H., M.R.S. and A. Klaesson wrote the scripts and developed pipelines for image analysis. D.R., D.K., C.S., A. Klaesson, A. Klemm, P.J.H. and O. Spjuth gave valuable suggestions on the use of software and creation of CellProfiler pipelines to analyze the MolBoolean data. D.R. and D.K. performed CellProfiler data analysis; P.J.H. and M.R.S. binned and plotted the data in graph format, and performed statistical analysis. A. Klaesson and J.H. contributed to the making of some figures. D.R., D.K., and O. Söderberg wrote the manuscript draft and prepared the figures. All authors contributed to the final version of the manuscript.

## Funding

## Competing interests

The authors declare the following competing interests: O. Söderberg is the inventor of the MolBoolean method, with patent number: US2022042069A1-2022-02-10. The patent is now held by Atlas Antibodies, therefore, as employees of Atlas Antibodies, T.M. and C.K. declare financial interest. The remaining authors declare no competing interests.
