## [Peer Review File · Nature Communications]

Reviewers' Comments:

Reviewer #1:

Remarks to the Author:

Summary

In their manuscript Raykova and Kerpatsou et al. describe a potential advancement of the proximity ligation assay (PLA), which is able to produce signals from free proteins (non-interacting) as well as from bound (interacting) proteins. While PLA is prone to detect crowding proteins as false-positive interactions due to the large distance which can be covered (~30-40 nm), it would represent an important advance to be able to quantify the abundance of both interaction partners investigated to estimate the risk of such false-positive events. Here, the authors describe a method they term "MolBoolean", which addresses this problem by generating two different fluorescent signals from the rolling circle amplification (RCA) of each protein individually, as well as double-positive signals from RCA of interacting proteins. While the idea and approach to solve this important problem of PLA is interesting, in our view the paper in its current state lacks sufficient benchmarking. Additionally, we are not quite convinced that the approach (two individual signals from single probes vs. dual signals of the same reporters from interacting probes) can sufficiently address the mentioned issue. However, this could be demonstrated by the authors but involves much more benchmarking and comparisons to state-of-the-art techniques.

General remarks

The detection of both, individual signals as well as interaction signals, is important and addresses a major problem in PLAs. However, we think the method presented here is not sufficiently benchmarked and therefore it remains uncertain whether MolBoolean represents a significant advance to solve this problem. Most notably, no thorough benchmarking against PLA and/or standard fluorescence microscopy with co-localization analyses is presented. Due to the fact that the interaction signal is spectroscopically not distinct from the co-localization of two individual signals, we are -at least not by the presented experiments- not convinced it can sufficiently address the problem. Our major and minor points including experimental suggestions are listed below. Most important in our view would be to prove that MolBoolean dual signals in fact originate from interacting proteins and a systematic comparison to PLA and/or standard co-localization analysis.

Major points

- 1) Benchmarking against negative sets is missing (systematic analysis of crowding proteins vs. interacting proteins and analysis of MolBoolean performance thereof); Authors should benchmark additional non-interacting proteins from different cellular compartments to gauge the background and applicability of the technique to other cellular compartments. This needs to be compared to state-of-the-art approaches (PLA, fluorescence co-localization microscopy)
- 2) Basic benchmarking that single signals are produced by non-interacting proteins, and dual signals originate from interactions is missing. The example demonstrated in Figure 2a is not sufficient since TGF-beta1 treatment obviously has a drastic effect not only on the interactions but also on cellular fitness and morphology of the cell population. Differentiation between co-localization of two signals produced from individual RCPs to dual signals originating from one RCP; e.g. using protein pairs with point mutations knowing to induce a loss of binding are warranted.
- 3) Benchmarking whether stoichiometries of individual proteins and interacting proteins are correctly determined is missing. Detected levels/ratios should be compared to published datasets or, ideally, compared to other techniques (e.g. cross-linking mass spectrometry).
- 4) While it is often warranted to present new findings applying a new method, which was not observable with state-of-the-art methods, we think it is not a must for a methods paper. However, it would be desired that the authors present a case where state-of-the-art methods were not able to e.g. differentiate crowding from interaction in the past or apply it to a different, previously unsolved (sensitivity) problem of PLA.

5) There is a heavy bias for nuclear proteins in the manuscript. Can the authors demonstrate that this method can be applicable to protein-protein interactions found in the cytoplasm, mitochondria, autophagosome, or endoplasmic reticulum?

Minor points

a) Kd of E-cadherin and B-catenin, does it fit to published literature? If not, discussion why it might be different should be included. If authors want to prove that MolBoolean can indeed quantify a change in affinity, this needs to be directly compared to data obtained with other methods.

b) How much of the free protein is due to non-specific signal? Can the author provide MolBoolean data on single protein target? And how this compares to when targeting dual proteins.

c) Can the authors demonstrate the specificity of MolBoolean with a cell line lacking the interaction partner? Such knockout experiments would strengthen the sensitivity and specificity assessment of the method.

d) Can the authors comment on the stoichiometries of proteins? Free vs. interacting protein populations in the different experiments. Do experiments where one interaction partner is highly abundant and the other less abundant result in a high-false interaction rate (e.g. due to crowding)?

e) First sentence of abstract: Determining levels of PPIs is not only essential for the analysis of signalling, but for many other things (e.g. assess effect of mutations, understand function of proteins and protein complexes, characterize disease, etc.)

f) The authors mention FRET in the introduction but don't elaborate on the difference and similarities to PLA or MolBoolean (e.g. regarding distance coverage, quantification of amounts of proteins involved in an interaction etc.). This should be added.

g) From the text and methods section it is not fully clear to us, which specific tags are used to generate the different signals (sequences for hybridization).

h) On page 4, lines 121-124 the authors report that the distance, which can be covered by the used size of affinity reagent and the length of the oligonucleotides is similar to what is reported for PLA. However, the authors do not demonstrate this.

i) On page 8, line 186 Table 1 with DNA design is mentioned but not included at this position in the manuscript (located at the end, page 16). Similarly, on page 8 line 194 Molecular Boolean Analysis seems to be missing.

j) Statistics between different conditions (Interaction, Protein A only, Protein B only) are missing for all RCPs/cell quantifications. From how many (individual) experiments was the data obtained?

Reviewer #2:

Remarks to the Author:

The manuscript by Raykova et al. described an interesting proximity-based imaging approach for the simultaneous profiling of protein interactions and singlet proteins. Essentially, this technique is an advanced in situ PLA, where DNA circle was cleaved by a single nickase in the presence of a single protein or a pair of nickases upon protein interaction. Barcodes were then inserted into the DNA circle and amplified by RCA. Despite this technique addresses a critical technical barrier in conventional proximity assays, i.e. false positives of close but non-interactive proteins, major revisions are needed to better illustrate and characterize this novel technology.

1) A major issue of the current manuscript is that the authors did not well describe the design of the novel set of DNA probes. This would be very difficult for readers who are not familiar with PLA assays. Great more details shall be given in Fig. 1, e.g. the affinity ligands, proteins, protein

complexes, and how they work together.

2) More details shall also be given for the design and optimization of the sequences for DNA probes. For example, how to design DNA probes, so that cleavage will only happen in the presence of the target protein. How to ensure dual cleavage events only occurs in the presence of the protein-protein complex. There is possibility that cleavage could occur even in the absence of the protein or protein-protein complex.

3) Other critical factors, such as the linkage between DNA and antibody, effect of the spacer length and sequence, etc shall also be detailed.

4) The authors also mentioned that the new design could also help evaluate the effectiveness of antibody-DNA conjugates. However, this was not clearly presented in the manuscript or the supplementary information. A cartoon shall be included to better illustrate how it works.

5) The goal is to address false positive issues caused by the co-localized but non-interactive proteins. Ideally, comparison with conventional techniques shall be performed to cross-validate this novel strategy.

Reviewer #3:

Remarks to the Author:

The authors present a method, MolBoolean, which uses a proximity-induced ligation and strand-invasion approach to detect free and bound proteins in fixed cells and tissues. This work builds on many studies that use antibody-oligonucleotide conjugates to detect proteins that are colocalized in fixed samples, however this approach uses two distinct oligonucleotide tags and strand invasion to determine whether a protein is present by itself (one tag invaded) or in a complex with another protein (two tags invaded), which theoretically could determine the amount of free and bound proteins in a cellular or tissue sample. The authors apply Molboolean to four different validated protein-protein complexes in cell culture and a tissue samples. In all of these cases, essentially a single experimental condition is shown with no controls, no statistical analysis, and no controls showing that relative changes in complex stoichiometry can be detected. Validation of these aspects of the method are necessary to confirm whether the detected interactions represent true positives and whether the relative quantities of 'free' and 'bound' species are at all accurate and within what dynamic range these species can be detected. Specific questions and comments are included below, but overall, more rigorous controls and quantitative/statistical analyses is needed to validate the conclusions of this study prior to publication.

1) The authors assert that the MolBoolean data collected on the B-catenin/E-cadherin interaction can be used to determine changes in affinity (i.e., K_d , as stated by the authors). First, the suggestion that dissociation constant (K_d) is being measured in these experiments is factually incorrect and misleading. The affinity between two proteins is an inherent property of the biophysical interaction that is not being measured by this method. What is being measured by this method (ideally) is the fraction of the two species that are in an AB complex and free (A and B separately), but this does not relate directly to the affinity between A and B, as there are many other factors involved in the formation of these complexes in cells that are not and cannot be accounted for. Relating the ratio of A, B and AB to the dissociation constant should be removed from the paper as the current language is confusing and not supported.

2) Controls (for example, absence of specific MolBoolean reagents, alternative targeting antibodies, oligo sequences) need to be included across the board in each condition of Figure 2. Specific delineation of details surrounding replicate measurements (biological and technical) and statistical comparisons are almost entirely missing and need to be included.

This approach proposes it has the advantage of measuring the concentrations (or relative amounts) of free proteins and those engaged in a bimolecular complex within the same experiment and therefore involves the competition of reagents for binding to complexes and free species, which could be present in quantities that vary by many orders of magnitude from one another. This scenario raises several questions.

3) First, how do you know that the concentration regime for the antibody-oligo complexes are in a

linear range for each of the different species, and not at the top or bottom of saturation regimes? Orthogonal measurements of free and bound species would need to be shown to confirm that absolute and relative quantities are indeed being accurately measured.

4) Additionally, for accurate, even relative, quantities of the free A and B and formed AB complexes to be measured in the same experiment, the relative quantities would need to all be somewhat near each other. This could significantly limit the approach, and the authors should present theoretical arguments as well as experimental data on different systems to convincingly show that these scenarios can be accounted for by MolBoolean.

5) The value of being able to measure the free and complexed populations of native proteins in a sample is being able to make quantitative comparisons across conditions, sample types, stimuli and other variables. However in all of the In the experimental systems presented a single 'static' condition was tested, which does not convincingly show that dynamics can be detected and quantified. The authors should run and include additional experiments wherein time-, dose- or condition-dependent changes in A, B and AB quantities can be visualized and matched to existing experimental data.

REVIEWER COMMENTS

Reviewer #1 (Remarks to the Author):

Summary

In their manuscript Raykova and Kermpatsou et al. describe a potential advancement of the proximity ligation assay (PLA), which is able to produce signals from free proteins (non-interacting) as well as from bound (interacting) proteins. While PLA is prone to detect crowding proteins as false-positive interactions due to the large distance which can be covered (~30-40 nm), it would represent an important advance to be able to quantify the abundance of both interaction partners investigated to estimate the risk of such false-positive events. Here, the authors describe a method they term “MolBoolean”, which addresses this problem by generating two different fluorescent signals from the rolling circle amplification (RCA) of each protein individually, as well as double-positive signals from RCA of interacting proteins. While the idea and approach to solve this important problem of PLA is interesting, in our view the paper in its current state lacks sufficient benchmarking. Additionally, we are not quite convinced that the approach (two individual signals from single probes vs. dual signals of the same reporters from interacting probes) can sufficiently address the mentioned issue. However, this could be demonstrated by the authors but involves much more benchmarking and comparisons to state-of-the-art techniques.

General remarks

The detection of both, individual signals as well as interaction signals, is important and addresses a major problem in PLAs. However, we think the method presented here is not sufficiently benchmarked and therefore it remains uncertain whether MolBoolean represents a significant advance to solve this problem. Most notably, no thorough benchmarking against PLA and/or standard fluorescence microscopy with co-localization analyses is presented. Due to the fact that the interaction signal is spectroscopically not distinct from the co-localization of two individual signals, we are -at least not by the presented experiments- not convinced it can sufficiently address the problem. Our major and minor points including experimental suggestions are listed below. Most important in our view would be to prove that MolBoolean dual signals in fact originate from interacting proteins and a systematic comparison to PLA and/or standard co-localization analysis.

Major points

1) Benchmarking against negative sets is missing (systematic analysis of crowding proteins vs. interacting proteins and analysis of MolBoolean performance thereof); Authors should benchmark additional non-interacting proteins from different cellular compartments to gauge the background and applicability of the technique to other cellular compartments. This needs to be compared to state-of-the-art approaches (PLA, fluorescence co-localization microscopy)

We recognize the importance of additional benchmarking and can see how it benefits the paper, which is why we have, in accordance with your recommendation, included several additional negative control experiments with non-interacting proteins that are abundantly expressed in different cellular compartments. In addition to the previously shown E-cadherin – Lamin A/C experiment (Fig. 3b) that targets proteins primarily located in the cell membrane and the nuclear lamina, we also performed a staining for GM130 – MT-CO1 (Suppl. Fig. 3c, d), which are markers for the Golgi apparatus and mitochondria respectively. We estimated the interaction background and found it comparable to the levels generated from *in situ* PLA. We also demonstrated that visually the signals localize to the correct compartments of the cell, and validated that with co-localization immunofluorescence staining

using the same pair of antibodies. Furthermore, we designed a padlock system (Suppl. Fig. 4a-e) which makes it impossible for dual-colored signals to be generated even from very closely positioned proteins in a complex, so any detection of such signals is an imaging-induced false positive. We were able to demonstrate a very low signal overlap when using the padlock system with the well-established interaction E-cadherin – β -catenin, which provides proof that the great majority of dual-colored signals in other assays are indeed veritable interactions. The artefacts induced by microscopy are negligible and very likely fixable with doing the image analysis in 3D. We have also compared all our MolBoolean assays to IF and *in situ* PLA stains, and have performed technical controls by omitting one or the other antibody from each pair, observing correlating results across methods.

2) Basic benchmarking that single signals are produced by non-interacting proteins, and dual signals originate from interactions is missing. The example demonstrated in Figure 2a is not sufficient since TGF-beta1 treatment obviously has a drastic effect not only on the interactions but also on cellular fitness and morphology of the cell population. Differentiation between co-localization of two signals produced from individual RCPs to dual signals originating from one RCP; e.g. using protein pairs with point mutations knowing to induce a loss of binding are warranted.

We agree that additional data supporting this was needed, so we addressed this point by using our padlock design which generates only individual signals (Suppl. Fig. 4), as well as by targeting β -catenin with two different antibodies (Fig. 2). These two experiments are set up so that they highlight the ability of MolBoolean to produce individual signals by non-interacting proteins (padlock assay) and dual signal from proximal epitopes (β -catenin – β -catenin assay). We would also like to note that in the case of TGF- β 1 treatment, in addition to its obvious effects on cell morphology, it also causes a change in binding affinity between E-cadherin and β -catenin (Fig. 5a). Our assay demonstrates that despite the increased abundance of these proteins per cell upon treatment, a lower fraction of the detected signals was dual-stained, and there was an increase in individual signals, in accordance with what we observe morphologically and in literature. We would like to thank Reviewer 1 for the excellent suggestion of including protein pairs with point mutations that induce a loss of binding. We have now included such an assay that is based on the use of cells which do not normally express E-cadherin. We performed MolBoolean on AGS cells stably transfected with expression vectors that contain either wild type E-cadherin, or E-cadherin with a mutation that interferes with efficient binding to β -catenin (V832M) (Fig. 4). We observed that when performing an E-cadherin – β -catenin stain in each transfected line, there was a marked difference in the level of individual and dual signals and a shift of mutant E-cadherin from the cell membrane to the cytoplasm.

3) Benchmarking whether stoichiometries of individual proteins and interacting proteins are correctly determined is missing. Detected levels/ratios should be compared to published datasets or, ideally, compared to other techniques (e.g. cross-linking mass spectrometry).

Detection of proteins with MolBoolean is dependent on the affinity of the antibodies used. Therefore, despite the great advantages that MolBoolean offers over other immunoassays, we realize and acknowledge that our method is semi-quantitative and is still subjected to some of the limitations of other techniques based on the utilization of antibodies, fluorescence and microscopy. We have extended the discussion to clarify this for the readers. Absolute quantification of protein levels is not possible with immuno-based techniques, MolBoolean included. However, what our method has demonstrated is an ability to detect relative levels and changes in these levels in different conditions. Therefore we feel that it is fair to compare MolBoolean to other methods of the same type, such as classical immunofluorescence (IF) (because similarly to MolBoolean it offers the possibility of detection of two proteins simultaneously and their visual localization to specific cell compartments) and *in situ* PLA (because it resembles MolBoolean's feature for detecting protein interactions based on proximity and produces quantifiable RCPs). This is why we complemented each one of our MolBoolean assays with a triplicate of IF and *in situ* PLA experiments performed with the same

antibodies in the same concentrations. Taken together, the data generated from IF and *in situ* PLA comes close to providing the same level of information as MolBoolean, but it still falls short for several reasons. Although IF shows the expression of the two (or more) proteins of interest in their respective compartments, it can only be measured in terms of fluorescence and does not allow for the quantification of discrete signals in the cell. In addition, the somewhat diffuse nature of IF staining can lead to wrongly interpreting closely positioned fluorescent signals as colocalizing. On the other hand, *in situ* PLA allows for the quantification of interactions on single-cell level and their localization, but does not offer any information on the levels of free proteins in the cell. Still, with the help of these two nowadays standard techniques, we were able to validate that the localization of MolBoolean signal in different cell compartments (including cell membrane, nucleus, endosomes, Golgi apparatus, mitochondria, etc) is congruent with literature and the results obtained via IF and *in situ* PLA. The amount of colocalizing signal detected with MolBoolean and *in situ* PLA was proportional, yet MolBoolean was able to detect a higher number of signals per cell which is closer to the absolute number of molecules expected in reality.

4) While it is often warranted to present new findings applying a new method, which was not observable with state-of-the-art methods, we think it is not a must for a methods paper. However, it would be desired that the authors present a case where state-of-the-art methods were not able to e.g. differentiate crowding from interaction in the past or apply it to a different, previously unsolved (sensitivity) problem of PLA.

Our TGF- β 1 treatment experiment (Fig. 5a) is a good example of how MolBoolean outperforms PLA. It is known that the binding of E-cadherin to β -catenin decreases upon treatment, however if one were to rely solely on PLA data, the opposite would appear to be true. This is because while the number of interacting E-cadherin and β -catenin molecules in each cell increases, so do the levels of each protein, and proportionally, the complex formation is actually decreased. This is possible to observe with MolBoolean, because it simultaneously detects both free and bound proteins and that allows for data normalization. With *in situ* PLA, we only observe increase in protein complexes after treatment, but do not get any information about the even bigger increase of free proteins.

5) There is a heavy bias for nuclear proteins in the manuscript. Can the authors demonstrate that this method can be applicable to protein-protein interactions found in the cytoplasm, mitochondria, autophagosome, or endoplasmic reticulum?

One reason why we focused on nuclear proteins was the fact that they are notoriously more difficult to stain compared to cytoplasmic and membrane-bound proteins, due to the presence of additional nuclear membrane that needs to be sufficiently permeabilized for reagents to penetrate it, so we found that using a challenging subcellular location is a good attestation of MolBoolean's performance. However we recognize that we had indeed neglected many other locations in the cell, and it is fair to demonstrate how our method works for those. To this end, we introduced several new assays: PDGFR β – Clathrin (membrane/ endosomes, Fig. 7b), PDIA3 – Calreticulin (endoplasmic reticulum/ cytoplasm, Fig. 7a), GM130 – MT-CO1 (Golgi complex/ mitochondria, Suppl. Fig. 3c, d), where we observed specifically localized stain with MolBoolean.

Minor points

a) K_d of E-cadherin and B-catenin, does it fit to published literature? If not, discussion why it might be different should be included. If authors want to prove that MolBoolean can indeed quantify a change in affinity, this needs to be directly compared to data obtained with other methods.

We understand and are fully aware that the measurement we offered in the initial version of the manuscript was relative and not the actual K_d, as detection and quantification of every protein molecule in the cell is not possible and is limited by the affinity of the antibodies used in MolBoolean.

However, we acknowledge that we didn't stress this enough, and in order to avoid confusion, we have now completely removed any mention of K_d from the paper.

b) How much of the free protein is due to non-specific signal? Can the author provide MolBoolean data on single protein target? And how this compares to when targeting dual proteins.

We have now complemented all our experiments with omitting controls – that is to say, we have performed MolBoolean using only one or the other primary antibody of the pair we use for the complete experiment, and we have kept the rest of the reagents unaltered. This set-up allows measuring any lack of specificity of the proximity probes that can cause false positive detection of free protein. To complement this further, our β -catenin/ β -catenin experiment (Fig. 2) that targets a single protein with two different antibodies is fit to show unspecific mono-colored signal, which in itself is determined by the specificity of each individual primary antibody (like all other antibody-based methods, MolBoolean is highly dependent on the high quality and specificity of the affinity reagents used, so it is only as good as the antibodies). Finally, our padlock experiment (Suppl. Fig. 4) is designed in a way that makes possible to determine the number of unspecific dual-colored signals, since only single-color signals are generated. When all this data is taken together, it gives a good idea of the unspecific background stain in every signal category that is to be expected.

c) Can the authors demonstrate the specificity of MolBoolean with a cell line lacking the interaction partner? Such knockout experiments would strengthen the sensitivity and specificity assessment of the method.

To address this point, we have compared two MolBoolean co-stainings against E-cadherin and β -catenin – in MCF7 cells (where both of these proteins are highly expressed), and in U2OS cells (which only express β -catenin, but not E-cadherin) (Fig. 3a). Our results demonstrate the specificity of MolBoolean in detecting the two partner proteins and their level of interaction in the double-positive line MCF7, and in detecting solely β -catenin and no interaction in U2OS cells.

d) Can the authors comment on the stoichiometries of proteins? Free vs. interacting protein populations in the different experiments. Do experiments where one interaction partner is highly abundant and the other less abundant result in a high-false interaction rate (e.g. due to crowding)?

This is an important topic, and we now have emphasized both in the Introduction and Discussion that neither MolBoolean, nor PLA or FRET can be used as means to determine physical interactions between proteins. What the MolBoolean method elucidates is whether, or not, the distance between the proteins of interest is below a certain threshold value.

e) First sentence of abstract: Determining levels of PPIs is not only essential for the analysis of signalling, but for many other things (e.g. assess effect of mutations, understand function of proteins and protein complexes, characterize disease, etc.)

We fully agree. Initially we formatted our study as a Brief Communication, which restricted us significantly when it comes to allowed word count, but now that we have expanded to a full article format, we were more than happy to correct this omission.

f) The authors mention FRET in the introduction but don't elaborate on the difference and similarities to PLA or MolBoolean (e.g. regarding distance coverage, quantification of amounts of proteins involved in an interaction etc.). This should be added.

Similar to the abstract, we were trying to keep a low word count that fit with the format of Brief Communication, but we fully agree that one can expand on the background of other similar methods.

We have now addressed this by expanding the Introduction section accordingly.

g) From the text and methods section it is not fully clear to us, which specific tags are used to generate the different signals (sequences for hybridization).

We regret that we have left this unclear, and kindly ask you to refer to Table 1 which lists all sequences used in MolBoolean. We use a naming system where any oligonucleotide sequences labelled “A” are meant to act in concert for the detection of signal of a general protein of interest named A. The same applies to any sequence named “B”. Therefore arm A, for example, is part of the proximity probe A that recognizes protein A, and tag A is the one incorporated in the circle using arm A as a template. The resulting RCP is then detectable by fluorescently labelled detection oligonucleotide A. The same goes for protein B, arm B, tag B and detection oligonucleotide B.

h) On page 4, lines 121-124 the authors report that the distance, which can be covered by the used size of affinity reagent and the length of the oligonucleotides is similar to what is reported for PLA. However, the authors do not demonstrate this.

The estimated distance both for *in situ* PLA and for MolBoolean is indeed based on the size of the affinity reagents and the oligonucleotide length, and is theoretical, which is why we haven't demonstrated it but just presented the logic used for making an estimate. We have now expanded on this further in the Introduction.

i) On page 8, line 186 Table 1 with DNA design is mentioned but not included at this position in the manuscript (located at the end, page 16). Similarly, on page 8 line 194 Molecular Boolean Analysis seems to be missing.

We apologize for the inconvenience but all figures and tables are located at the end of the manuscript instead of wherever first mentioned, in accordance with the submission requirements of Nature Communications. If the paper is accepted for publication, this positioning will be changed.

j) Statistics between different conditions (Interaction, Protein A only, Protein B only) are missing for all RCPs/cell quantifications. From how many (individual) experiments was the data obtained?

We have now performed all assays, including omitting controls, in triplicate. In addition, three images were taken per experiment every time, and then instead of averaging the detected RCPs over the number of counted cells per image, we performed our analysis on single-cell level. The distribution of RCP levels across the cells are now shown in box and whisker plots, and statistical data is available in the figure legends and the Source Data file for all experiments that required comparison between conditions.

Reviewer #2 (Remarks to the Author):

The manuscript by Raykova et al. described an interesting proximity-based imaging approach for the simultaneous profiling of protein interactions and singlet proteins. Essentially, this technique is an advanced *in situ* PLA, where DNA circle was cleaved by a single nickase in the presence of a single protein or a pair of nickases upon protein interaction. Barcodes were then inserted into the DNA circle and amplified by RCA. Despite this technique addresses a critical technical barrier in conventional proximity assays, i.e. false positives of close but non-interactive proteins, major revisions are needed to better illustrate and characterize this novel technology.

1) A major issue of the current manuscript is that the authors did not well describe the design of the novel set of DNA probes. This would be very difficult for readers who are not familiar with PLA assays. Great more details shall be given in Fig. 1, e.g. the affinity ligands, proteins, protein complexes, and how they work together.

We understand that for readers unfamiliar with proximity-based assays, and even for some readers familiar with those, the complex MolBoolean design may be difficult to grasp with the provided explanations, therefore we extended the Results section with much more detail in terms of how the whole design works together and expanded Fig. 1 accordingly. We also added a sub-section in Supplementary Notes (Oligonucleotide Design), in which we elaborate on the design of all MolBoolean components and in what way it is important for the system to function. We feel that these additional explanations together with the revised and improved existing figure make the principle much more understandable.

2) More details shall also be given for the design and optimization of the sequences for DNA probes. For example, how to design DNA probes, so that cleavage will only happen in the presence of the target protein. How to ensure dual cleavage events only occurs in the presence of the protein-protein complex. There is possibility that cleavage could occur even in the absence of the protein or protein-protein complex.

We acknowledge the necessity to elaborate on our design and make clearer to the reader how it works. Therefore, we added a detailed sub-section in Results called Principle of the MolBoolean method, and a sub-section in Supplementary Notes called Oligonucleotide Design, where we provide further extensive clarifications as to how the designs were conceived, in what way their sequences and folding contribute to their function, and why cleavage is specific and cannot occur in the absence of the protein. We also demonstrated the specificity of hybridization between oligos and the enzyme specificity of MolBoolean in an *in solution* experiment shown in Suppl. Fig. 1.

3) Other critical factors, such as the linkage between DNA and antibody, effect of the spacer length and sequence, etc shall also be detailed.

We agree that it is important for the readers to understand the factors essential for the method. We have hence added a section with oligonucleotide design in Supplementary and extended the Introduction and Discussion parts to clarify this. We have also included experiments on primary antibody conjugates, using different conjugation chemistries to show alternative approaches (Suppl. Fig. 5b). We have also added a section describing *in solution* tests to validate the specificity of the enzymatic steps (Suppl. Fig. 1).

4) The authors also mentioned that the new design could also help evaluate the effectiveness of antibody-DNA conjugates. However, this was not clearly presented in the manuscript or the supplementary information. A cartoon shall be included to better illustrate how it works.

It was our intention to point out the potential of MolBoolean for use as an antibody validation tool, as in the case of the β -catenin double-stain we performed using two antibodies against the same antigen (Fig. 2). This assay demonstrates how well two antibodies against two different epitopes within the same protein bind to their targets. If either one or both of them are cross-reactive, there will be less dual signal generated by the MolBoolean system. However, one should also remember to consider the fact that the efficiency of creating dual signal also depends on the level of epitope occupancy and efficiency of the enzymatic steps, i.e., how efficiently the tags get incorporated. We have extended the Discussion with a section regarding this.

5) The goal is to address false positive issues caused by the co-localized but non-interactive proteins. Ideally, comparison with conventional techniques shall be performed to cross-validate this novel strategy.

We have extended the Introduction and Discussion to emphasize for the readers that MolBoolean, similarly to other methods, measures colocalization of proteins within certain distance, but we cannot make a claim that these proteins interact for sure. We agree that cross-validation with conventional techniques was warranted and have now performed co-stains with MolBoolean, *in situ* PLA and IF for almost all of our experiments. In addition, we performed an assay where a wild type version of E-cadherin or E-cadherin with a point mutation affecting its binding to β -catenin were transfected into a line that does not normally express E-cadherin but expresses β -catenin (Fig. 4). This way we created an environment where the two proteins are both abundant but interaction is only detected between WT E-cadherin and β -catenin, and not where the mutant version of E-cadherin was expressed. These results were also supported by *in situ* PLA data in a previous publication by Figueiredo et al., *Eur J Hum Genet* 2013 Mar; 21(3): 301–309 (referenced in the text).

Reviewer #3 (Remarks to the Author):

The authors present a method, MolBoolean, which uses a proximity-induced ligation and strand-invasion approach to detect free and bound proteins in fixed cells and tissues. This work builds on many studies that use antibody-oligonucleotide conjugates to detect proteins that are colocalized in fixed samples, however this approach uses two distinct oligonucleotide tags and strand invasion to determine whether a protein is present by itself (one tag invaded) or in a complex with another protein (two tags invaded), which theoretically could determine the amount of free and bound proteins in a cellular or tissue sample. The authors apply Molboolean to four different validated protein-protein complexes in cell culture and a tissue samples. In all of these cases, essentially a single experimental condition is shown with no controls, no statistical analysis, and no controls showing that relative changes in complex stoichiometry can be detected. Validation of these aspects of the method are necessary to confirm whether the detected interactions represent true positives and whether the relative quantities of ‘free’ and ‘bound’ species are at all accurate and within what dynamic range these species can be detected. Specific questions and comments are included below, but overall, more rigorous controls and quantitative/statistical analyses is needed to validate the conclusions of this study prior to publication.

1) The authors assert that the MolBoolean data collected on the B-catenin/E-cadherin interaction can be used to determine changes in affinity (i.e., K_d , as stated by the authors). First, the suggestion that dissociation constant (K_d) is being measured in these experiments is factually incorrect and misleading. The affinity between two proteins is an inherent property of the biophysical interaction that is not being measured by this method. What is being measured by this method (ideally) is the fraction of the two species that are in an AB complex and free (A and B separately), but this does not relate directly to the affinity between A and B, as there are many other factors involved in the formation of these complexes in cells that are not and cannot be accounted for. Relating the ratio of A, B and AB to the dissociation constant should be removed from the paper as the current language is confusing and not supported.

We are thankful to Reviewer 3 for rightfully pointing out the issues causing confusion and lack of clarity. We understand and are fully aware that the measurement we offered in the initial version of the manuscript was relative and not the actual K_d , as detection and quantification of every protein molecule in the cell is not possible and is limited by the affinity of the antibodies used in MolBoolean.

However, we acknowledge we did not stress that fact enough, and in order to avoid misunderstanding or misleading claims, we have now completely removed any mention of K_d from the paper in accordance with your suggestion.

2) Controls (for example, absence of specific MolBoolean reagents, alternative targeting antibodies, oligo sequences) need to be included across the board in each condition of Figure 2. Specific delineation of details surrounding replicate measurements (biological and technical) and statistical comparisons are almost entirely missing and need to be included.

We have now expanded the range of biological and technical controls, and we have provided more refined single-cell analysis, as well as statistics (where applicable) for all our experiments. Omitting of reagents was done in the course of method optimization, and the *in solution* experiment (Suppl. Fig. 1) which we included in this latest manuscript version demonstrates the effect of some reagents' omission or swapping. Furthermore, we performed omitting controls in which we used either one of the primary antibodies alone and kept all other reagents for the majority of assays.

This approach proposes it has the advantage of measuring the concentrations (or relative amounts) of free proteins and those engaged in a bimolecular complex within the same experiment and therefore involves the competition of reagents for binding to complexes and free species, which could be present in quantities that vary by many orders of magnitude from one another. This scenario raises several questions.

3) First, how do you know that the concentration regime for the antibody-oligo complexes are in a linear range for each of the different species, and not at the top or bottom of saturation regimes? Orthogonal measurements of free and bound species would need to be shown to confirm that absolute and relative quantities are indeed being accurately measured.

We try to use high antibody concentrations to saturate as many epitopes as possible (excess of antibodies will be removed by washing). We have extended the discussion section with a paragraph describing how the results of the analysis would be affected by alterations in antibody concentrations and enzyme activity. We hope that this will make it clear to the readers that the balance between free and interacting proteins recorded cannot be used as an absolute measurement of levels of these molecules in a cell. This is equally true for other methods based on recording proximal binding of antibodies, such as FRET and PLA, in which detection of interaction/proximal events require that both epitopes in a protein complex are bound by probes.

In the manuscript, we do try to clearly convey the message that MolBoolean cannot provide absolute measurements, but can measure changes in the amounts of free and proximal proteins using the same experimental settings. We cannot compare different antibodies, since they have different affinities and are differently affected by, for example, length of washes and harshness of the buffers used. However, we feel that by including side-by-side *in situ* PLA and IF co-stains, we have provided solid basis for comparison.

4) Additionally, for accurate, even relative, quantities of the free A and B and formed AB complexes to be measured in the same experiment, the relative quantities would need to all be somewhat near each other. This could significantly limit the approach, and the authors should present theoretical arguments as well as experimental data on different systems to convincingly show that these scenarios can be accounted for by MolBoolean.

We have extended the discussion to clarify that the MolBoolean quantifications are relative. To demonstrate that MolBoolean can be used also for situations where levels of the targets vary, we have included examples where the expression of one protein is reduced or not expressed. The MolBoolean method demonstrates a lot of flexibility in accounting for different scenarios, for example when

showing the decrease in E-cadherin – β -catenin binding and the increased levels of free E-cadherin upon TGF- β 1 stimulation compared to control conditions (Fig. 5a). Furthermore, in our PDIA3 knock-down experiment (Fig. 7a), we managed to decrease the levels of PDIA3 with 92.5%, i.e. the relative quantities of this protein were significantly different before and after silencing, which was accurately reflected by MolBoolean. In extreme conditions, where one of the supposed interaction partners was completely missing, such as in our experiment in which we stained for E-cadherin and β -catenin in two different cell lines, one of which does not express E-cadherin (Fig. 3a), MolBoolean had no problem detecting only the free, highly abundant β -catenin in U2OS cells and no E-cadherin, whereas it detected high levels of interaction of these two proteins in MCF7 cells. In addition, we performed an assay where a wild type version of E-cadherin, or E-cadherin with a point mutation affecting its binding to β -catenin were transfected into a line that does not normally express E-cadherin but expresses β -catenin (Fig. 4). This way we created an environment where the two proteins are both abundant but interaction is only detected between WT E-cadherin and β -catenin, and not where the mutant version of E-cadherin was expressed.

5) The value of being able to measure the free and complexed populations of native proteins in a sample is being able to make quantitative comparisons across conditions, sample types, stimuli and other variables. However in all of the In the experimental systems presented a single ‘static’ condition was tested, which does not convincingly show that dynamics can be detected and quantified. The authors should run and include additional experiments wherein time-, dose- or condition-dependent changes in A, B and AB quantities can be visualized and matched to existing experimental data.

We appreciate this suggestion and in order to address it, we have included a number of experiments. We have the TGF- β 1 treatment assay (Fig. 5a), in which cells change morphology, adhesion properties and levels of free and bound E-cadherin and β -catenin upon prolonged stimulation; the Clathrin – PDGFR β assay (Fig. 7b), where cells were treated for a short time with a different ligand producing the opposite result (increased interaction); as well as our knock-down assay (Fig. 7a) where we used siRNA to silence PDIA3 and observed the effects of that on the free protein and its levels of binding to Calreticulin. In all these experiments we demonstrated that MolBoolean can detect and quantify proteins in different dynamic conditions.

Reviewers' Comments:

Reviewer #1:

Remarks to the Author:

Summary

Raykova and Kerpatsou et al. have performed several additional experiments and thoroughly revised their previous version of the manuscript. It now includes systematic comparison of the different analyses to in situ PLA and immunofluorescence (IF). Additionally, the authors have included the analysis of additional protein pairs and controls, including the investigation of an interaction with a missense mutation previously shown to disrupt binding. This data and the application of the padlock system clearly demonstrating that the signals from interaction RCPs and signals from individual RCPs are clearly distinct, does convince that "MolBoolean" performs as wanted.

Importantly, the authors have now also analysed their different scenarios (single protein targeting (Fig 2), interaction vs. no-interaction (Fig 3), wild-type vs. missense mutation (Fig 4), ligand-inducible interaction (Fig 5), nuclear interactions (Fig 6), dynamic conditions (Fig 7) from three independent experiments and performed quantitative assessment including statistical analyses of their data.

I very much appreciate the revised manuscript and consider MolBoolean a significant advancement over current state of the art techniques. The ligand-inducible interaction between E-cadherin and beta-catenin nicely demonstrates the advantage over in situ PLA, where the absence of a signal from non-interacting proteins leads to misinterpretation of the result.

The validation and relative quantification of protein-protein interactions under endogenous conditions, including in tissue sections, is very important for basic research as well as clinical use. Therefore, in situ PLA is currently a widely used technique. MolBoolean significantly improves this important assay type. I recommend the manuscript for publication. Below are my minor remarks for consideration.

Minor remarks:

Abstract

- "a novel method to identify interactions" (MolBoolean is not shown to identify interactions, rather validation; also compare with line 87 and title)
- "reports the levels" (MolBoolean does not report levels of proteins) -> indicates relative abundances

Intro

- "cellular signalling activity" (protein-protein interactions are not only relevant for signalling activity)
- Line 70, FRET range (can you provide a reference?)

Supplementary Figure 1

- Label marker bands/indicate sizes, label product bands etc. (e.g. nicked)

Discussion

- Lines 342: "this distance is similar to what has been reported for in situ PLA" (I agree that this is likely, but no distance assessment is demonstrated (e.g. using steric linkers, epitope mapping, or similar). Therefore, please consider rephrasing this sentence.)

Answer to major point #2

- "We would also like to note that in the case of TGF- β 1 treatment, in addition to its obvious effects on cell morphology, it also causes a change in binding affinity between E-cadherin and β -catenin (Fig. 5a)." (I want to emphasize once more that the observed effect on E-cadherin and beta-catenin upon TGF-beta1 treatment should not be described as "a change in binding affinity". This statement would include assays/experiments, which are able to determine binding affinities. In PLA (and/or MolBoolean) and similar methods, one does not measure binding affinity but observes more or less complexes. Whether this is a result of a change in affinity, localization, etc

is not known and should therefore not be stated.)

Reviewer #2:

Remarks to the Author:

The authors have adequately addressed my previous comments and concerns in the revised manuscript and I believe it is ready to be accepted for publication in its current form.

Reviewer #3:

Remarks to the Author:

In the revised version of MolBoolean the authors have presented new experimental evidence to support the validity and utility of their method. These changes include a greater textual explanation of the method itself, including specifics surrounding the antibody-oligonucleotide conjugates used, tag-invasion (NOT/AND) workflow, statistics and application to additional protein pairs. At the heart of comments made by all three reviewers were questions about whether the data presented support the notion that MolBoolean is capable to detecting bona fide protein complexes and non-complexed copies of those proteins. The extra data and textual revision by the authors further supports their manuscript, but there remain issues with the presentation and interpretation of key data that need clarification in order to support the claims in the manuscript.

1) The authors respond to several reviewers that they have included key controls where components of the molboolean proximity complexes are omitted, non-interacting proteins are interrogated and the complement of free and interacting proteins are assessed across different abundance regimes. All of these data are important to establish that molboolean can indeed robustly detect A, B and AB complements. These data are mostly absent from the revision, however, and should be explicitly presented in main figures.

2) The authors state that they ability to quantify the amount of free and complexed proteins helps normalize data. How? In all of the data presented the authors report on the RCP/cell for the dual-invaded signal and the individual signals (complexed and free). How is this data useful and integrated as presented? And since the authors point out in their manuscript and response that like PLA and FRET, MolBoolean cannot determine the fraction of proteins in a complex, what does this method add that cannot be accomplished by in situ PLA?

3) The key data presented in Fig 4 and Fig 5 showing that proteins that contain a mutation reducing their propensity to interact (Fig 4) and cell treatment to change complex dynamics (Fig 5) appear to have been done in very different biological settings. It is conspicuous that in both experiments where there is higher BCat-Ecad complex formation the cells are very tightly packed, but in the chosen imaging fields where there is lower complex formation (presented to be introduced by mutation or TGFB treatment) the cells are dispersed. These experiments and results are not convincing, then, that the method is indeed picking up specific differences in complex formation rather than abundance, as cell density and contact are known to affect things like E-cadherin localization, interactions and modifications.

Reviewer #4:

Remarks to the Author:

The authors of the manuscript title Molboolean: A Boolean analysis of protein-protein interactions at a molecular level have address all comments and provided additional high-quality studies which have strengthen the design and applicability of their technique.

The major comments concerning basic benchmarking were very well addressed with a repertoire of sound experiments. All minor points were also sufficiently addressed, in particular the clarification of Kd and its omission from the manuscript.

Overall, the revised manuscript supports the major conclusion and claims presented by the

authors. The results clearly demonstrate the specificity and versatility of MolBoolean in assessing protein-protein interactions in different cellular compartments, under signaling-dependent conditions, and different cell types and tissues. It also demonstrates its advantages over in situ PLA and IF, in particular in being able to distinguish between free and in-complex protein molecules.

I recommend this revised manuscript for publication.

REVIEWER COMMENTS

Reviewer #1 (Remarks to the Author):

Summary

Raykova and Kerpatsou et al. have performed several additional experiments and thoroughly revised their previous version of the manuscript. It now includes systematic comparison of the different analyses to in situ PLA and immunofluorescence (IF). Additionally, the authors have included the analysis of additional protein pairs and controls, including the investigation of an interaction with a missense mutation previously shown to disrupt binding. This data and the application of the padlock system clearly demonstrating that the signals from interaction RCPs and signals from individual RCPs are clearly distinct, does convince that “MolBoolean” performs as wanted. Importantly, the authors have now also analysed their different scenarios (single protein targeting (Fig 2), interaction vs. no-interaction (Fig 3), wild-type vs. missense mutation (Fig 4), ligand-inducible interaction (Fig 5), nuclear interactions (Fig 6), dynamic conditions (Fig 7) from three independent experiments and performed quantitative assessment including statistical analyses of their data.

I very much appreciate the revised manuscript and consider MolBoolean a significant advancement over current state of the art techniques. The ligand-inducible interaction between E-cadherin and beta-catenin nicely demonstrates the advantage over in situ PLA, where the absence of a signal from non-interacting proteins leads to misinterpretation of the result.

The validation and relative quantification of protein-protein interactions under endogenous conditions, including in tissue sections, is very important for basic research as well as clinical use. Therefore, in situ PLA is currently a widely used technique. MolBoolean significantly improves this important assay type. I recommend the manuscript for publication. Below are my minor remarks for consideration.

Minor remarks:

Abstract

- “a novel method to identify interactions” (MolBoolean is not shown to identify interactions, rather validation; also compare with line 87 and title

Sentence changed to “...a novel method to detect interactions between endogenous proteins in various subcellular compartments, utilizing antibody-DNA conjugates for identification and signal amplification”.

- “reports the levels” (MolBoolean does not report levels of proteins) -> indicates relative abundances

“Reports the levels” changed to “indicates relative abundancies”.

Intro

- “cellular signalling activity” (protein-protein interactions are not only relevant for signalling activity)

Sentence entirely removed, as the next sentence elaborates on other aspects of relevance in addition to signaling activity.

- Line 70, FRET range (can you provide a reference?)

Reference added.

Supplementary Figure 1

- Label marker bands/indicate sizes, label product bands etc. (e.g. nicked)

Marker band sizes added, relevant product bands labeled.

Discussion

- Lines 342: “this distance is similar to what has been reported for in situ PLA” (I agree that this is likely, but no distance assessment is demonstrated (e.g. using steric linkers, epitope mapping, or similar). Therefore, please consider rephrasing this sentence.)

Sentence rephrased to “In the current secondary antibody-based design this theoretical distance is similar to what has been reported for in situ PLA.”

Answer to major point #2

- “We would also like to note that in the case of TGF- β 1 treatment, in addition to its obvious effects on cell morphology, it also causes a change in binding affinity between E-cadherin and β -catenin (Fig. 5a).” (I want to emphasize once more that the observed effect on E-cadherin and beta-catenin upon TFG-beta1 treatment should not be described as “a change in binding affinity”. This statement would include assays/experiments, which are able to determine binding affinities. In PLA (and/or MolBoolean) and similar methods, one does not measure binding affinity but observes more or less complexes. Whether this is a result of a change in affinity, localization, etc is not known and should therefore not be stated.)

We sincerely thank Reviewer #1 for their positive comments and appreciation of our revised manuscript. We were happy to honor the textual modifications suggested above. Regarding major point #2, our phrasing was indeed not precise in this case. We agree that speaking of binding affinity, as if it were possible to provide exact values for that, is incorrect. What we meant to say instead was along the lines of “observable changes in protein complex formation”. To make sure we avoid confusion in the readers, we have modified the manuscript text accordingly.

Reviewer #2 (Remarks to the Author):

The authors have adequately addressed my previous comments and concerns in the revised manuscript and I believe it is ready to be accepted for publication in its current form.

We are happy to have addressed all concerns raised by Reviewer #2, and thank them for the constructive criticism that led to improving of the revised manuscript.

Reviewer #3 (Remarks to the Author):

In the revised version of MolBoolean the authors have presented new experimental evidence to support the validity and utility of their method. These changes include a greater textual explanation of the method itself, including specifics surrounding the antibody-oligonucleotide conjugates used, tag-invasion (NOT/AND) workflow, statistics and application to additional protein pairs. At the heart of comments made by all three reviewers were questions about whether the data presented support the notion that MolBoolean is capable to detecting bona fide protein complexes and non-complexed copies of those proteins. The extra data and textual revision by the authors further supports their manuscript, but there remain issues with the presentation and interpretation of key data that need clarification in order to support the claims in the manuscript.

1) The authors respond to several reviewers that they have included key controls where components of the molboolean proximity complexes are omitted, non-interacting proteins are interrogated and the complement of free and interacting proteins are assessed across different abundance regimes. All of these data are important to establish that molboolean can indeed robustly detect A, B and AB complements. These data are mostly absent from the revision, however, and should be explicitly presented in main figures.

We completely agree with the notion that the multiple control experiments we performed are important and necessary in order to demonstrate the robustness and reliability of the results obtained with the MolBoolean method, and need to be easily accessible. The reason they were not shown in the first version of the manuscript was because we initially had prepared it as a Brief Communication for Nature Methods and were pressed for space. The manuscript was then transferred to Nature Communications. In hindsight, it would have been better to have changed it to an article format at that stage. We apologize for any inconvenience this may have caused. Now that we have expanded it to full article, we still used the maximum number of display items allowed, but we agree that it is reasonable to demonstrate the control experiments in a better way. To correct this, we have now greatly expanded Fig. 2 (our experiment in which two β -catenin antibodies raised in different species were used against different epitopes of β -catenin) by including an extended view of the full MolBoolean experiment, displaying also technical controls omitting either of the two primary antibodies, comparisons with *in situ* PLA with omitting controls, quantifications for all of these assays, as well as comparison to immunofluorescent staining. Additionally, we have transferred what was previously Supplementary Fig. 4 (i.e. the padlock experiment) to

the main article text as Fig. 4. To keep within the limits allowed by the journal, we have regrouped some other figures.

2) The authors state that their ability to quantify the amount of free and complexed proteins helps normalize data. How? In all of the data presented the authors report on the RCP/cell for the dual-invaded signal and the individual signals (complexed and free). How is this data useful and integrated as presented? And since the authors point out in their manuscript and response that like PLA and FRET, MolBoolean cannot determine the fraction of proteins in a complex, what does this method add that cannot be accomplished by *in situ* PLA?

We normalized the data by dividing the number of RCPs we obtained for each category (A, B or AB) by the total number of all RCPs detected in that particular cell, and went on to plot the resulting fractions (noted as normalized RCPs/cell in the graphs and corresponding figure legends). This normalization could not be performed for *in situ* PLA data from the same experiment, as there is only one category of signal that PLA produces (which would correspond to the AB category in MolBoolean), and therefore the resulting fraction would always equal 1. Notably, whenever we performed normalizations, our goal was not to obtain a percentage that has biological relevance in each signal category, but rather to offer a fairer condition of comparison. In the normalized experiments, for various reasons there were differences between the two compared groups of cells analyzed, and we normalized simply to account for these differences, so that they do not lead to skewed conclusions. For example, in the TGF- β 1 treatment experiment, the treated cells are much larger than the controls and as such contain more proteins. To look at the ratios of proteins in each category and how they change upon treatment is, in our view, more valuable than to simply observe how many signals were detected per cell. We do not speculate to say that the obtained percentages depict with precision the absolute percentages of molecules of each category (A, B, AB) in the cell – much like our other quantifications, these are relative measurements, but shown in a way that in our opinion aids the correct interpretation of the result.

3) The key data presented in Fig 4 and Fig 5 showing that proteins that contain a mutation reducing their propensity to interact (Fig 4) and cell treatment to change complex dynamics (Fig 5) appear to have been done in very different biological settings. It is conspicuous that in both experiments where there is higher BCat-Ecad complex formation the cells are very tightly packed, but in the chosen imaging fields where there is lower complex formation (presented to be introduced by mutation or TGF β treatment) the cells are dispersed. These experiments and results are not convincing, then, that the method is indeed picking up specific differences in complex formation rather than abundance, as cell density and contact are known to affect things like E-cadherin localization, interactions and modifications.

Within the frame of one experiment, cells were seeded in equal numbers and grown in identical conditions, unless a treatment was indicated (which would then be the only difference in conditions). In both the E-cadherin mutation experiment and the TGF- β 1 treatment, we therefore had the same number of cells per well between control and “condition” cells and same growth time before fixation. In the first case, mutant cells were

expressing a pathological form of E-cadherin typical of gastric cancer. It has been previously shown that the V832M mutation in the β -catenin binding site of E-cadherin causes cells that express the aberrant protein to not aggregate as they normally would (see Figueiredo et al. 2013, which in the manuscript is reference 19). In the case of TGF- β 1 stimulation, it is well established that TGF- β 1 induces EMT, which, among other effects, increases cell migration and has a role in tumor metastasis (e.g. see Ikushima & Miyazono, 2010 or Liarte et al. 2020, i.e. references 49 and 20 respectively). In concurrence with literature, we also observed that, either as an effect of the V832M mutation that interferes with the formation of adherens junctions, or as a result of the prolonged TGF- β 1 stimulation, cells lose their typical morphology and instead become larger, stretching out over a bigger surface, and tend to migrate away from one another, as opposed to control cells which stay close together and in contact with one another (there are similar images shown in Liarte et al, 2020). When we performed any of our experiments, we did so in triplicate, and within one iteration we would always acquire a minimum of three images per condition. Each image has the same parameters, including a field of view with a set area. That is why, even though the number of seeded cells in each category was the same, since their size and motility were different, less of the enlarged and spread-out cells could be captured per image. It was therefore no accident that our non-control images depict less cells, but rather a direct consequence of the biological processes taking place. We aimed to capture our images from random spots on the slides, and even though not every single non-control image appears homogeneously spread out, we chose to show the most representative ones, in which it was clear that the treatment worked. Nonetheless, all images were analyzed together and the shown quantifications include all data.

To avoid potential confusion in the readers, we have addressed the Reviewer's point by expanding certain parts of Results or Discussion and adding additional explanations and references to clarify why cells appear as they do.

We thank Reviewer #3 for their comments, which have improved the clarity and quality of our manuscript, and hope that we have now sufficiently addressed all of their concerns.

Reviewer #4 (Remarks to the Author):

The authors of the manuscript title Molboolean: A Boolean analysis of protein-protein interactions at a molecular level have address all comments and provided additional high-quality studies which have strengthen the design and applicability of their technique.

The major comments concerning basic benchmarking were very well addressed with a repertoire of sound experiments. All minor points were also sufficiently addressed, in particular the clarification of Kd and its omission from the manuscript.

Overall, the revised manuscript supports the major conclusion and claims presented by the authors. The results clearly demonstrate the specificity and versatility of MolBoolean in assessing protein-protein interactions in different cellular compartments, under signaling-

dependent conditions, and different cell types and tissues. It also demonstrates its advantages over in situ PLA and IF, in particular in being able to distinguish between free and in-complex protein molecules.

I recommend this revised manuscript for publication.

We are very grateful to Reviewer #4 for their positive comments and are glad that they were pleased with how we addressed their previous critique.

Reviewers' Comments:

Reviewer #3:

Remarks to the Author:

The authors have addressed key outstanding issues with additional text and changes to their figure presentation and therefore this manuscript should be published as is.